# The EMC acts as a chaperone for membrane proteins

Carolin J. Klose [1,2,5], Kevin M. Meighen-Berger [1,5], Martin Kulke [1], Marina Parr[3], Barbara Steigenberger[4], Martin Zacharias[1], Dmitrij Frishman[3] & Matthias J. Feige [1] ✉

Structure formation of membrane proteins is error-prone and thus requires chaperones that oversee this essential process in cell biology. The ER membrane protein complex (EMC) is well-defined as a transmembrane domain (TMD) insertase. In this study, we characterize an additional chaperone function of the EMC. We use interactomics and systematic studies with model proteins to comprehensively define client features for this EMC chaperone mode. Based on this data, we develop a machine learning-based tool for client prediction. Mechanistically, our study reveals that the EMC engages TMDs via its EMC1 subunit and modulates their orientation within the lipid bilayer. Productive TMD assembly reduces binding to the EMC chaperone site. Taken together, our study provides detailed insights into an EMC chaperone function, further establishing the role of the EMC as a multifunctional molecular machine in membrane protein biogenesis.

Membrane proteins transport matter and information through biological lipid bilayers and are thus indispensable for any living cell. In eukaryotes, membrane proteins are encoded by around one quarter of their genes[1,2]. The biosynthesis of most eukaryotic membrane proteins occurs at the endoplasmic reticulum (ER), where this structurally and functionally diverse class of proteins is integrated into the lipid bilayer and acquires its native structure[1,2]. Recent studies have shown that several mechanisms exist by which transmembrane domains (TMDs) of a membrane protein are inserted into the ER lipid bilayer, which constitutes an early defining step in their structure formation. As examples, the heterotrimeric Sec61 translocon allows co-translational insertion of typical hydrophobic TMDs into the ER membrane via its lateral gate[3]. For tail-anchored (TA) proteins, which possess a single C-terminal TMD, two different routes have been described. The guided entry of tail-anchored proteins (GET) complex caters to more hydrophobic TMDs[4], whereas the ER membrane protein complex (EMC) can insert less hydrophobic TMDs of TA proteins[5]. Beyond this role as an insertase for TA proteins, the EMC can also integrate N-terminal TMDs with their N-terminus facing the ER/extracellular space ($N_{exo}$)[6] as well

as C-terminal TMDs of a $C_{exo}$ orientation that are not succeeded by any long sequences which would be required for co-translational insertion via Sec61[7]. The EMC is thus a key protein when it comes to integrating terminal TMDs into the lipid bilayer and is a defining factor in shaping membrane protein topology[8]. This insertase function of the EMC depends on its Get1/YidC homologous subunit EMC3[9,10], which is a member of the Oxa1 superfamily of insertases. However, EMC3 is only one of the ten EMC subunits in human cells, which raises the question of why the EMC is so much more complex than other Oxa1 family members[8]. Several studies suggest that, from yeast to man, the multisubunit EMC has additional roles beyond its insertase function. This is also reflected by the loss of many membrane proteins in EMC knockout studies beyond likely insertase clients[11,12]. These include, e.g. the ABC transporter Yor1 in yeast (homologous to human CFTR)[13], the gap junction protein Connexin 32[14], rhodopsin[15–17], and many others[11,12,18–20]. Furthermore, the link of the EMC to ER-associated degradation (ERAD) factors[21] and its strong association with ER stress[22] may also suggest additional roles beyond its insertase function. Such roles could include a chaperone function in which the EMC would stabilize incompletely

[1]Department of Bioscience, TUM School of Natural Sciences, Center for Functional Protein Assemblies (CPA), Technical University of Munich, Garching, Germany. [2]Department of Molecular Machines and Signaling, Max Planck Institute of Biochemistry, Martinsried, Germany. [3]Department of Bioinformatics, Wissenschaftszentrum Weihenstephan, Technical University of Munich, Freising, Germany. [4]Mass Spectrometry Core Facility, Max Planck Institute of Biochemistry, Martinsried, Germany. [5]These authors contributed equally: Carolin J. Klose, Kevin M. Meighen-Berger. ✉e-mail: Matthias.Feige@tum.de

folded/assembled membrane proteins until their biogenesis is completed[11]. Many membrane proteins possess non-ideal TMDs, including length mismatch to the ER lipid bilayer or exposed polar residues—features which are functionally needed but compromise stability in the membrane[1,23]. Generally, these features become alleviated during folding, assembly, and transport—but require chaperoning during biosynthesis in the ER. Since the EMC directly integrates TMDs and additionally participates in a large insertase complex[24], it is necessarily in contact with yet incompletely assembled TMDs, which would ideally position it for such a TMD chaperone function. Recent structures of the EMC support the idea that the EMC has more than one functional site in the membrane. These structures show that the EMC possesses two potential substrate engagement sites in the membrane, with its insertase function located on one side of the complex, and a lipid-filled cavity on the other side[25–28]. This lipid-filled cavity is made up of the EMC subunits 1, 3, 5, and 6 and appears to be accessible from the ER membrane[25–28]. A recent structure of the EMC in complex with a calcium channel assembly intermediate[29], as well as molecular insights into different EMC functions[7,11], support the idea of the EMC being a multifunctional molecular machine in membrane protein biogenesis. Mechanistic insights into a putative chaperone function of the EMC, however, are missing. Our study addresses this open issue in eukaryotic membrane protein biogenesis. We show that the EMC can recognize membrane proteins with signatures of incomplete structure formation to chaperone and safeguard their biogenesis.

## Results

### The EMC interacts with a diverse set of membrane proteins that do not classify as insertase clients

The molecular architecture of the nine-subunit human EMC can be divided into three layers. It shows a transmembrane core, mainly formed by subunits EMC3 through EMC6, a cytoplasmic cap comprising EMC2 along with either EMC8 or EMC9, and a characteristic ER lumenal part formed by subunits EMC1, EMC7, and EMC10[26–28]. A key element within the transmembrane core is a gated cavity, which has been assigned to the insertase function of the EMC[25–28,30,31]. Importantly, on the other side of the transmembrane core lies a second cavity that is continuous with the lipid bilayer and mainly formed by EMC1 (Fig. 1a, b). In cryo-EM structures of the EMC, this cavity has been found to bind lipids (highlighted in Fig. 1b) and was therefore designated as "lipid-filled cavity"[26]. This prominent molecular element within the EMC has been suggested to be involved in membrane protein binding[26], but mostly remains uncharacterized. Since the EMC supports the biogenesis of many membrane protein clients that cannot be explained by its insertase action, we focused on this lipid-filled cavity as the second characteristic feature of the EMC within the membrane to gain a better understanding of the diverse molecular functions of the EMC.

To analyze the interactome of the EMC, including enrichment of transient interaction partners of the lipid-filled cavity, we used site-specific crosslinking in HEK 293T cells coupled with mass spectrometry (MS). Incorporation of the unnatural amino acid 4-Benzoyl-phenylalanine (Bpa), which contains a photoactivatable crosslinker moiety, was performed at 17 different positions within EMC1, all lining the lipid-filled cavity (Supplementary Fig. 1a). Mutants were designed using experimental structures and an Alphafold 2 multimer model of the human EMC (Fig. 1b and Supplementary Fig. 1b). Subsequently, the crosslinking efficiency to endogenous interaction partners was evaluated via immunoprecipitation (IP) of crosslinked adducts under stringent washing conditions followed by immunoblot analyses (Supplementary Fig. 1c). Based on these analyses, three positions within EMC1 that showed high crosslinking efficiency and were well distributed across the lipid-filled cavity were selected for a further mass spectrometric analysis: at the brace/crossbar (R513), at the lumenal side of the single EMC1 TMD (D961), and at its cytoplasmic side (R981)

(Fig. 1b and Supplementary Fig. 1c, d). In the IP-MS analyses, 506 interacting proteins were significantly enriched in the Bpa crosslinked samples over the control condition, with over 200 of these being transmembrane proteins, including both single-pass and multipass membrane proteins (Fig. 1c, Supplementary Data 1). A subset of these hits was selected to validate for binding to the endogenous EMC by co-IP and western blot (Supplementary Fig. 2a, b). Further analysis of the interacting transmembrane proteins revealed that the majority of them did not classify as known EMC insertase clients, which are low-hydrophobicity TA proteins, signal anchor and multipass TM proteins with $N_{exo}$ topology, as well as multipass TM proteins with $C_{exo}$ topology and a short C-terminus (see Fig. 1c and Supplementary Data 1). Roughly 50% of the identified single-pass TM proteins possess a signal peptide and are thus expected to be inserted into the ER via the Sec61 translocon rather than the EMC, and only one-third possess terminal transmembrane helices that serve as signal or tail anchor (see Fig. 1c and Supplementary Data 1). Similarly, approximately two-thirds of the multipass TM proteins we identified have a predicted $N_{cyto}$ orientation, implying that the EMC is not responsible for the insertion of their N-terminal TMDs. These findings indicated that the EMC can interact with a diverse set of membrane proteins beyond its insertase clients, including quality control factors, but also potential chaperone clients. Our interactomics data suggest that these may be as simple as single-pass transmembrane proteins. To directly test this, isolated TMDs of six different interaction partners, from single-pass to multipass transmembrane proteins, were queried for their EMC interaction. To analyze single TMD interactions with the EMC, we fused them to superfolder GFP (sfGFP), equipped them with a classical N-terminal ER-targeting signal sequence, and first verified their correct membrane integration (Supplementary Fig. 2c, d and Supplementary Table 1). For the selected correctly integrated TMDs, we found in co-IP experiments that they interacted with the EMC, although to a different degree (Fig. 1d and Supplementary Fig. 2e). The identification of this quite heterogeneous class of EMC interactors prompted us to further investigate common elements or motifs that unite these diverse proteins.

### Defining EMC's intramembrane client binding preferences

Studying intramembrane chaperone interactions in a systematic manner poses unique challenges. In particular, when observing interactions with multipass transmembrane proteins, it is often unclear which TMD interacts with the chaperone of interest and whether cytoplasmic or lumenal domains further contribute to the interaction. Additionally, naturally occurring proteins are likely to contain redundant chaperone recognition motifs for robustness. To overcome these inherent complexities of natural clients, we employed a recently developed minimal transmembrane protein model system[32] to define the binding preferences of the EMC. This minimal consensus membrane protein (ConMem) is a type I transmembrane protein with a classical ER signal sequence and is thus subject to Sec61-mediated insertion. It encompasses a consensus TMD sequence derived from ~200 single-pass plasma membrane proteins and is characterized by its stability, natural intracellular transport behavior, and the absence of chaperone recognition sites (Fig. 2a)[32]. The transmembrane domain of ConMem thus serves as an inert scaffold, allowing site-specific amino acid substitutions and experimental assessment of their effects on EMC binding. Notably, our data show that the EMC can bind individual TMDs of natural membrane proteins (Fig. 1d). By exchanging the central amino acid of ConMem (valine at position 13) with all 19 possible other amino acids and normalizing binding to the reference (V13), we assessed the EMC binding behavior in a quantitative manner (Fig. 2b and Supplementary Fig. 3a). Our findings indicate that charged and polar residues within a TMD generally increased EMC binding, albeit to a different extent. To analyze more of the transmembrane domain of these

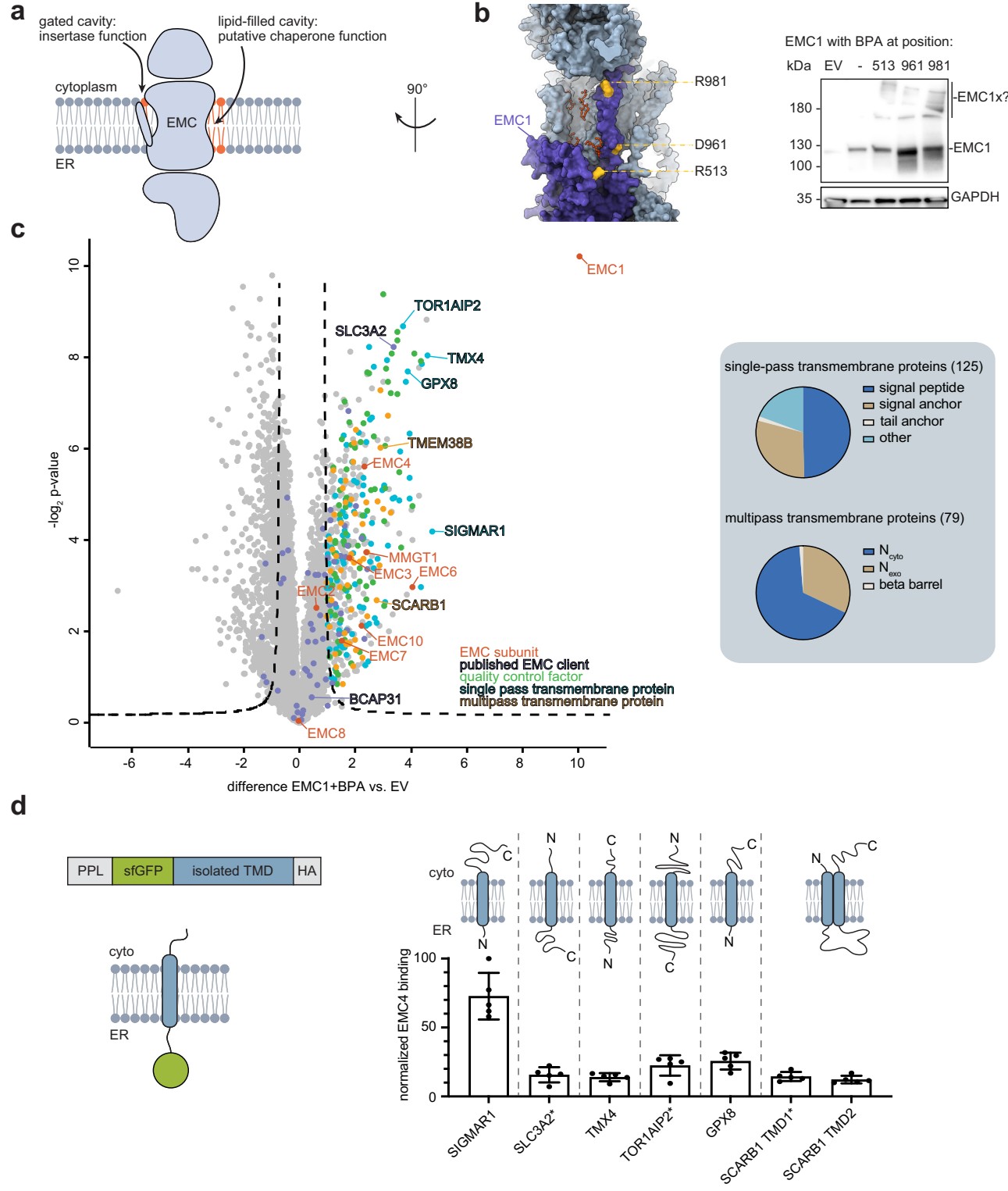

minimal EMC clients and delineate positional preferences, eight selected residues (G, S, Y, P, E, D, K, R) that demonstrated strong binding or were of interest due to their physicochemical properties (e.g. helix-breaking residues G and P and charged residues E and D) were shifted to various positions within the ConMem TMD (Fig. 2c and Supplementary Fig. 3b, c) and their correct membrane integration was verified (Supplementary Fig. 3d and Supplementary Table 1). Interaction analyses on this extended ConMem set revealed different positional preferences for different amino acids and generated a comprehensive profile of EMC binding to individual TMDs

(Supplementary Table 2), which was based on both amino acid properties and their respective positions within the TMD.

**Proteome-wide prediction of EMC chaperone substrate binding**
Our comprehensive set of binding data (Supplementary Table 2) revealed a broad range of interactors for the EMC, as expected for a molecular chaperone. Even more importantly, this dataset allowed us to transition from analysis to prediction and develop a machine learning approach capable of computing binding affinities for the EMC on a proteome-wide scale (Fig. 3a). We termed this multilayer

**Fig. 1 | A broad variety of transmembrane proteins and ER quality control factors interact with the EMC. a** The EMC possesses two distinct cavities in the intramembrane space: a gated cavity for insertion of TM helices and a lipid-filled cavity on the opposite side of the complex. **b** View of the lipid-filled cavity in an AlphaFold 2 model of the human EMC, overlaid with bound lipids (red) resolved in cryo-EM studies (PDB 7ADO[26]). EMC1 is colored in purple. Positions lining this cavity that were chosen for Bpa incorporation and subsequent photocrosslinking and mass spectrometry are highlighted in yellow. Crosslinks formed upon UV irradiation are visualized on an anti-EMC1 immunoblot of total cell lysates. EV empty vector. **c** The interactome of EMC1 was determined by Bpa crosslinking at the lipid-filled cavity followed by IP-MS. The data displays an average over three positions (R513, D961, R981, each individually replaced by Bpa). EMC1 interacts with a large number of ER quality control factors (green), previously described EMC clients (violet), and numerous single-pass (aquamarine) and multipass TM proteins

(orange). Proteins selected for further validation and characterization are labeled with the respective gene name. Interacting TM proteins were further classified according to the number and relative positioning of their TMDs (right). **d** The EMC binds to isolated TMDs of interacting membrane proteins. Isolated TMDs of interactors from **c** were inserted into a model protein as depicted in the schematic (left), which contains a preprolactin ER-targeting sequence (PPL), an sfGFP moiety, and an HA tag and is inserted into the ER membrane in a type I topology. The schematics above the graph represent the architecture of the proteins from which the TMDs were derived. Note that in some constructs the sequence of the isolated TMD was inverted to preserve the orientation it would have in its original protein (indicated with an asterisk), thus it only partially reflects the natural TMD. Binding to the endogenous EMC (in comparison to empty vector) was assessed by co-IP and normalized to the highest measured value ($n = 5$, mean ± standard deviation (SD)).

perceptron neural network *i*ntramembrane binding *pred*ictor for the *EMC* or short ipred[EMC]. Using the dataset obtained by ConMem binding studies, 40 position-specific features within the AAindex[33,34] (see Supplementary Table 3; e.g., hydrophobicity index, localized electrical effect, and others) were identified to correlate with EMC binding. When applying ipred[EMC] to a small test set of 15 ConMem sequences, which were part of our binding data but separated from the training dataset of 44 ConMem sequences, a robust correlation between predicted values and experimental findings was observed (Supplementary Fig. 4a). To test the model's predictive performance on independent data, we evaluated its predictions on a subset of the proteomic data shown in Fig. 1c (non-membrane proteins, EMC subunits, mitochondrial proteins, and single-pass proteins were excluded). In a simplified approach, we always scored a multipass TM protein with the highest score found for one of its TMDs (see the "Methods" section for details). We found that proteins identified as EMC binders by mass spectrometry indeed exhibited higher predicted EMC affinity scores (Supplementary Fig. 4b). Although our ipred[EMC] training set was limited, these analyses demonstrate that the model's predictions were consistent with independent experimental measures of EMC binding. To further validate the EMC binding predictor, it was employed to design TMD sequences in the context of a poly-leucine background with varying degrees of EMC binding (weak, moderate, and strong). After verifying their correct integration into the ER membrane (Supplementary Fig. 4c and Supplementary Table 1), we experimentally assessed EMC binding and found it to correlate excellently with the predictions (Fig. 3b and Supplementary Fig. 4d). It should be noted, however, that a bias for arginine residues was present, due to its strong impact on binding in the context of isolated TMDs (Fig. 2b, c). Using our validated ipred[EMC], we applied it to a set of previously reported EMC clients to identify TMDs within these proteins that possess a high predicted EMC binding affinity (Supplementary Data 2, Fig. 3c). Although the extent of binding did vary from the computed affinities, we found that for all three proteins tested the TMDs which ipred[EMC] predicted to bind to the EMC indeed did (Fig. 3c and Supplementary Fig. 4e). The outcome of these further validation experiments (Fig. 3b and c) was used for a second iteration of ipred[EMC] trained on the dataset of 77 ConMem sequences, 20 of which were allocated to the test set (Supplementary Table 4). This refined version of the algorithm was further adapted to accommodate transmembrane domains of varying lengths, facilitating the analysis of endogenous client interactions (see the "Methods" section for details).

This improved version of ipred[EMC] was used to uncover EMC binding signatures on a membrane proteome-wide level. To this end, transmembrane sequences of all human membrane proteins were scored with ipred[EMC] on a scale of 0–100. Like before, for simplification, the highest-scoring TMD within each protein was deemed representative of the overall protein's score. Multipass TM proteins exhibited a higher average normalized score of 20.5 ± 5.1 compared to 13.1 ± 3.9 for single-pass proteins (Fig. 3d), possibly due to a higher

statistical likelihood of one TMD among many to be a good binder. In general, the binding score correlated with the overall number of TMDs in a protein (Supplementary Fig. 4f).

An analysis of the top 500 membrane proteins with the highest predicted EMC affinity unveiled that nearly half belonged to the category of transporters (47.6%), encompassing, e.g. ion channels (16.4%), while the second most prevalent class comprised transmembrane signal receptors such as G protein-coupled receptors (GPCRs), together accounting for 19.5% of the analyzed cohort (Fig. 3e). Transporters are prime examples for proteins whose TMDs are weakly hydrophobic, often containing polar or charged amino acids to allow the transport of polar/charged solutes through the membrane. Frequently, such TMDs are only marginally stable in the lipid bilayer and may only be stabilized in the context of a fully assembled protein, likely requiring chaperoning during the assembly process.

## EMC1 engages transmembrane domains via hydrophobic interactions in the lipid bilayer

Our data suggest a direct interaction between EMC and the TMDs of diverse membrane proteins. Within the EMC, EMC1 is most likely to be involved in this interaction, as suggested by the structure of the lipid-filled cavity (Fig. 1b), our Bpa crosslinking mass spectrometry data (see Fig. 1c), and a recent cryo-EM structure of an ion channel assembly intermediate bound to the EMC[29]. However, the EMC1 TMD does not possess a clearly identifiable feature, such as a negatively charged patch within the membrane, that would allow us to explain our observation that lysine, and even more so arginine, within the ConMem TMD promote binding (see Fig. 2), making the binding mode unclear. Thus, to test for a direct ConMem:EMC1 interaction and gain insights into a possible binding mode, we introduced Bpa at different positions within the single ConMem TMD and analyzed crosslinking to different subunits of the EMC (Fig. 4a and Supplementary Fig. 5a). Surprisingly, we detected crosslinks between EMC1 and ConMem only for a Bpa site on the opposite face of the central, binding-promoting R13 (Fig. 4a), arguing against a direct EMC1:R13 engagement. To better understand this unexpected finding, we performed molecular dynamics (MD) simulations of the EMC in an ER membrane-like lipid environment where the TMD of ConMem R13 was initially positioned at the lipid-filled cavity in a similar position as the first TMD of the ion channel subunit in the cryo-EM structure of the client-bound EMC[29]. In agreement with our Bpa crosslinking data, the simulations showed that R13 favors an orientation facing away from EMC1 into the hydrophobic lipid bilayer, placing amino acid 19 of ConMem, for which we had observed crosslinking, in an adequate position to interact with EMC1 (Fig. 4b). Of note, throughout the simulation, the ConMem helix adopted a tilted conformation in the lipid bilayer, with R13 pointing towards the charged lipid headgroups and only the ER-proximal part of its TMD contacting the TMD of EMC1 (Supplementary Movies 1 and 2). In agreement with this finding, we also observed lipid molecules diffusing between EMC1 and ConMem R13 in the

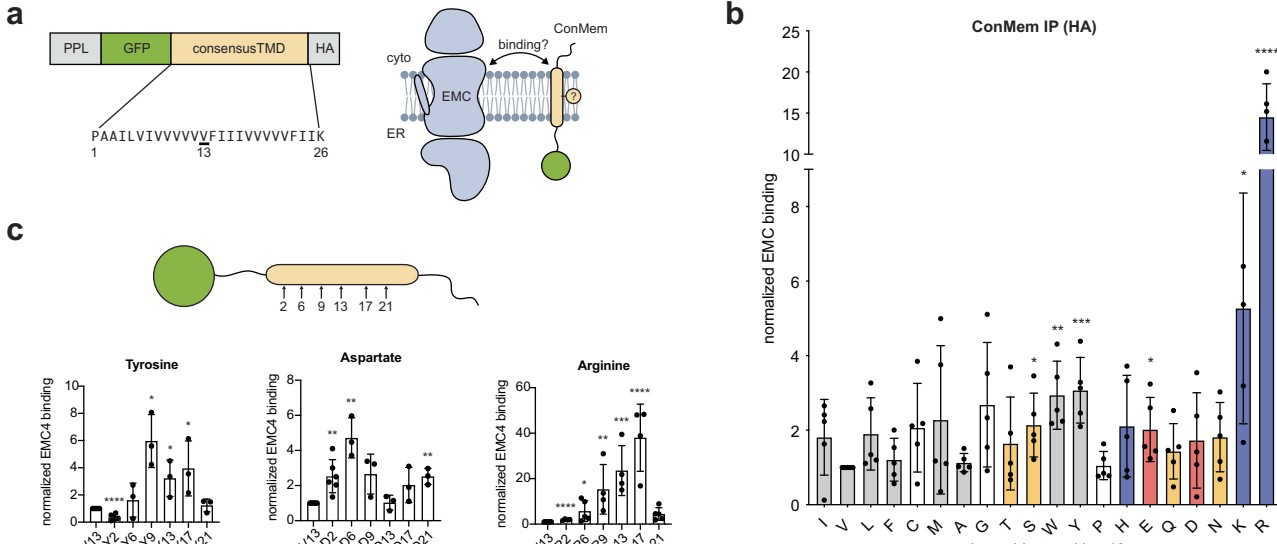

**Fig. 2 | EMC interaction with a TMD depends on its molecular composition.**
**a** The minimal consensus membrane protein "ConMem" (left) is a model protein as described in ref. 32. Systematically changing its inert TMD sequence allows analyses of intramembrane recognition processes as depicted in the schematic on the right. For a first set of EMC binding experiments (shown in **b**), the central valine at position 13 (underlined) was exchanged. **b** Systematic analysis of the interaction of ConMem variants with the endogenous EMC in HEK 293T cells. The central valine at position 13 was replaced with all other 19 amino acids, and binding to endogenous EMC was assessed by co-IP against the HA-tag on ConMem. Residues analyzed are ordered according to the Kyte–Doolittle scale for hydrophobicity[61] and color-coded according to physicochemical properties (gray: hydrophobic/aromatic,

yellow: polar, red: negatively charged, blue: positively charged). Binding was assessed in five independent replicates and normalized to the original consensus sequence (V at position 13) (mean ± SD, *P-value < 0.05, **P-value < 0.01, ***P-value < 0.001, ****P-value < 0.0001, two-tailed Student's t tests). **c** Positional binding dependency of the EMC to ConMem variants with substitutions at indicated positions throughout the ConMem TMD. Binding was assessed in at least three independent replicates by co-IP against the HA-tag on ConMem and normalized to the original consensus sequence (V at position 13) (mean ± SD, *P-value < 0.05, **P-value < 0.01, ***P-value < 0.001, ****P-value < 0.0001, two-tailed Student's t tests).

cytoplasmic leaflet of the membrane (Supplementary Fig. 5b). In contrast to our expectations, but in agreement with the Bpa crosslinking, the MD simulations thus did not predict a direct EMC1:R13 interaction, but rather a hydrophobic interaction site between the ER-lumenally oriented ends of the ConMem and EMC1 TMDs (Fig. 4b). To validate the MD simulations, we introduced mutations at the positions within EMC1 which were predicted to form the most contacts with residues of ConMem throughout the simulations (V963, S966, S967, F970, F974) (Supplementary Fig. 5c). Among these, certain variants were identified that did not substantially disturb the integrity of the EMC (Supplementary Fig. 5d) but notably diminished the binding of ConMem (Fig. 4c and Supplementary Fig. 5e), in agreement with the predicted interaction interface and mode of substrate binding. Subsequently, to cross-validate the observed effects in silico, the binding-deficient EMC1 mutants were simulated with ConMem R13, and the mode of TMD binding was indeed altered (Supplementary Fig. 5c), further corroborating the predicted interaction interface. Notably, residues previously proposed to be important for chaperone client engagement by the EMC[29] did not affect ConMem binding (Supplementary Fig. 5e, f). Additionally, we introduced mutations at the EMC1 brace as we found this to be in close proximity to ConMem in our simulations. These variants either negatively impacted the overall EMC stability or did not affect ConMem binding, thus providing no clear evidence for the brace to be involved in client engagement (Supplementary Fig. 5d–f). To further validate the unexpected binding mode, we exchanged hydrophobic residues in ConMem R13, which were found to be part of the simulated interaction interface, to more polar threonines. Again, in agreement with our crosslinking experiments, MD and EMC1 mutational data, mutants at the ER-lumenally oriented end of the ConMem TMD reduced binding to EMC1, while more distal mutants did not (Fig. 4d and Supplementary Fig. 6a). As a further validation of our structural model of EMC1:client binding, we performed pairwise

site-specific cysteine crosslinking in cells. We observed crosslinks between ER-proximal cysteines in the ConMem TMD and a cysteine at position 963 in EMC1, with the highest efficiency of crosslinking observed for residues 5 and 7 in ConMem. This further corroborates our structural model in which the distance of these two residues from Val 963 (around 10 Å) is in agreement with the BMH crosslinker spacer length of 13 Å (Fig. 4e and Supplementary Fig. 6b). Taken together, our combined experimental and computational data suggest that EMC1 binds chaperone clients in the membrane via its TMD and that binding is focused towards the ER-proximal side of the EMC1 TMD and is mediated by mostly hydrophobic contacts. Our data furthermore suggests that a charged residue within a client TMD that induces binding to EMC does not necessarily directly engage EMC1 but rather can position a TMD in an adequate geometry to engage EMC1.

Having defined the detailed binding mode of EMC1 to ConMem as a model TMD, we sought to investigate whether this also translated to more complex natural membrane proteins and what effects a lack of EMC binding would have on these. To address these questions, we focused on a diverse set of membrane proteins, containing one to seven TMDs, that we previously identified to bind to the EMC when performing crosslinking at the lipid-filled cavity (Fig. 1). For these proteins, mutating the putative intramembrane client binding site in EMC1 (Fig. 4c) in most cases reduced binding to the EMC (Fig. 4f). To globally assess a role of the EMC for these clients, we transiently knocked down EMC1 and assessed the effects on endogenous levels of all of these clients for which we could validate commercially available antibodies. For SIGMAR1, BCAP31, and TMEM38B, we observed significantly reduced levels under EMC1 knockdown conditions (Fig. 4g and Supplementary Fig. 6c). Taken together, this data shows that a broad panel of clients engages the EMC via a conserved binding site in EMC1 and that the stability of such proteins depends on the EMC.

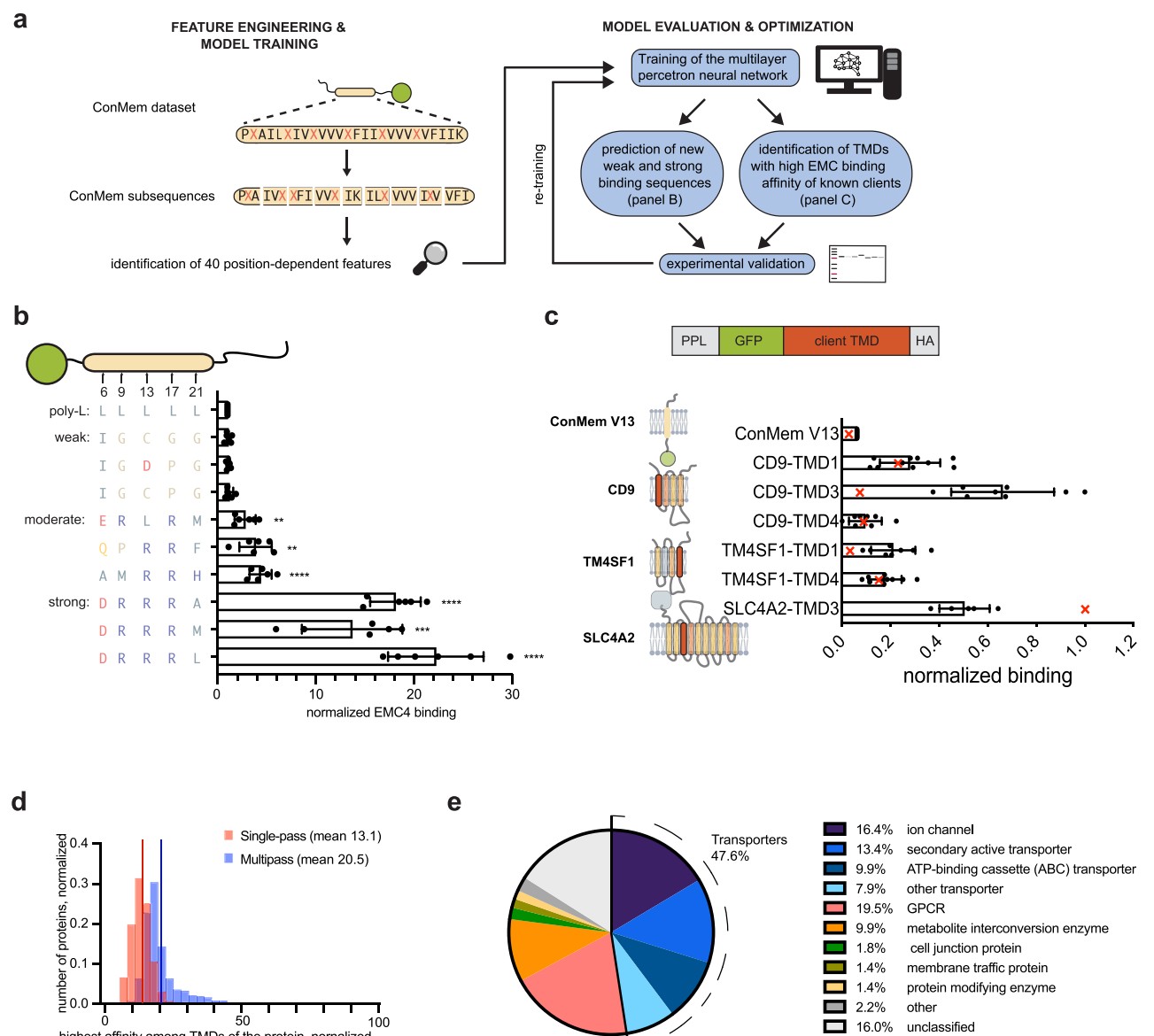

**Fig. 3 | A neural network allows to predict binding of transmembrane domains to the EMC. a** Architecture of the EMC binding predictor ipred^EMC. The neural network was trained with a dataset derived from 44 ConMem sequences (Fig. 2 and Supplementary Fig. 3), which were split into 9 subsequences each, from which features from the AAindex database were extracted. The predictor was then applied to design new sequences with different EMC affinities as well as to discover TMDs with high affinity within a list of known EMC clients. Experimental data from these two validation strategies served as a basis for a second iteration of training. This refined version of the predictor was ultimately applied to all TMDs of the human proteome to uncover new EMC chaperone clients. **b** Prediction and experimental validation of sequences with defined EMC interaction strength (strong/moderate/weak) integrated into a poly-leucine TMD. In a model protein (see Fig. 1d) containing a TMD composed of 24 leucines, flanked by proline and lysine as in ConMem (Fig. 2a), amino acid exchanges were performed at the positions indicated in the schematic based on predictions by ipred^EMC. The chosen weak binders were predicted to be the three most weakly-binding sequences, while the chosen strong binders were the three sequences with the highest predicted binding score. The three moderate binders were chosen from sequences with intermediate predicted binding scores. Binding of the EMC to these constructs was assessed by co-IP experiments in at least five independent replicates and normalized to the poly-leucine sequence (top) (mean ± SD, *P-value < 0.05, **P-value < 0.01, ***P-value < 0.001, ****P-value < 0.0001, two-tailed Student's t tests). **c** Application of the

predictor to previously reported EMC clients (Supplementary Data 2) to predict binding affinities for the individual transmembrane domains. Schematics of the selected proteins depict the number and orientation of their TMDs and are colored according to the predicted EMC affinity in shades of yellow (low) and red (high). Experimental validation of the predicted binding for a selection of client transmembrane domains is shown as black data points. Binding was assessed in at least five independent replicates and normalized to the highest obtained value. For comparison, the predicted values were also normalized on a 0 to 1 scale and plotted as red crosses. Please note that the topology of SLC4A2 is not well defined, and we show the one consistent with Uniprot annotation here, yet we cannot exclude a different orientation of the TMDs, which would affect our findings. **d** The ipred^EMC predictor was applied to the whole human membrane proteome to reveal the distribution of EMC binding signatures in all human membrane proteins. For multipass TM proteins, the highest-scoring TMD within the protein was used as the score for a given protein. Predicted affinities were compared between single-pass and multipass membrane proteins, or between $N_{cyto}$ and $N_{exo}$ membrane proteins. The number of proteins was normalized to the total number of proteins in their respective subset, and the affinity values were normalized to 0–100. **e** Analysis of the top 500 hits of the ipred^EMC predictor according to protein class using the PANTHER tool[62]. The shares of different types of transporters were added together, and these classes are labeled in shades of blue.

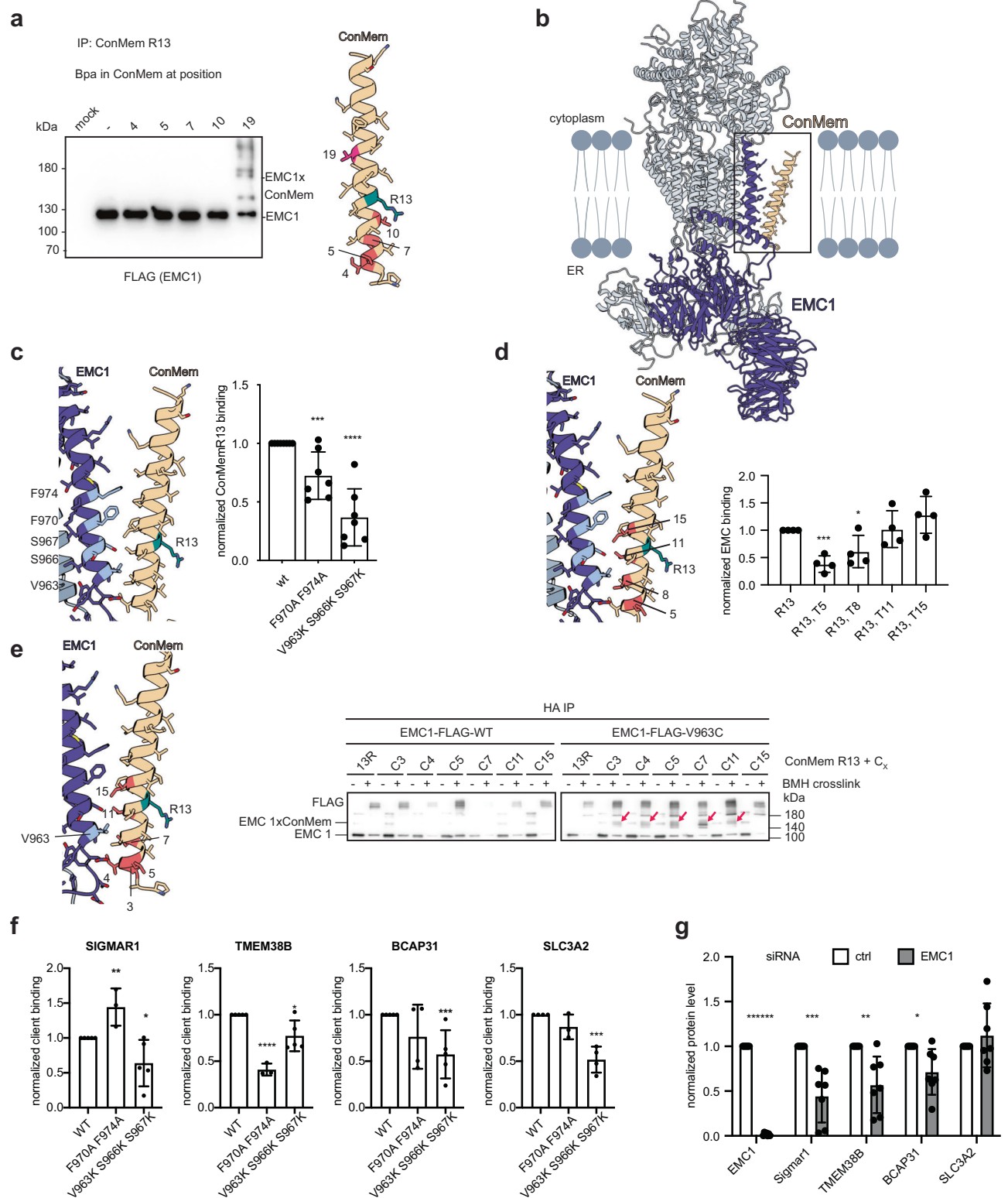

## Binding to the EMC modulates transmembrane domain orientation

Our analyses show that EMC1 binds to client TMDs via a mostly hydrophobic, ER-proximal site within the membrane. This raises two main questions: First, why are certain TMDs bound and others not, although TMDs in general are hydrophobic? And second, what consequences does binding have for the bound TMD and the EMC? To address these questions, we extended our experimentally validated

MD simulations on EMC1 client binding. For TMDs that we had observed to strongly bind the EMC, including ConMem R13, Y9, and D6 (see Fig. 2c), we observed a tilted orientation within the membrane in the absence of the EMC (Fig. 5a and Supplementary Fig. 7a; Supplementary Movie 1). Binding to the EMC led to a more upright orientation within the membrane (Fig. 5a and Supplementary Fig. 7a; Supplementary Movie 2). In contrast, for the very weak EMC binders ConMem V13, P9, and F13, we did not observe such a reorientation, and they

**Fig. 4 | Molecular characterization of EMC1 intra-membrane client engagement. a** Site-specific replacement of indicated residues within the transmembrane domain of a strong-binding ConMem variant (R13) with Bpa. Photocrosslinking at position 19 leads to the formation of covalent adducts with EMC1. Samples were immunoprecipitated against ConMem (HA-tag) and blotted for EMC1 (FLAG-tag). The ConMem TMD is shown on the right with amino acids individually replaced by Bpa, colored in red, position 19, resulting in the productive crosslink is highlighted in dark pink, and R13 in teal. **b** Molecular dynamics simulation of the interaction between EMC1 and ConMem R13 reveals a prevalent binding mode: the ConMem transmembrane domain is slightly tilted in the lipid bilayer with the R residue positioned towards the charged lipid headgroups of the lumenal leaflet, pointing away from the EMC1 TMD (EMC1 shown in dark blue). The observed interaction with the EMC is mainly mediated by hydrophobic residues. One representative pose of 5 replicates is shown. **c** Analysis of the EMC1 chaperone site by mutagenesis (residues colored in light blue) reveals the importance of hydrophobic interactions between EMC1 and ConMem R13. Binding was assessed by co-IP against myc-tagged EMC1 and immunoblotting against the HA-tagged ConMem R13 in at least seven independent replicates and normalized to EMC1 wt (mean ± SD, ***P*-value < 0.001, ****P*-value < 0.0001, two-tailed Student's *t* tests). **d** Exchange of hydrophobic residues in ConMem R13 corroborates the importance of hydrophobic interactions with EMC1. Mutation of ConMem residues to a more polar amino acid (threonine) reduced binding if close to the predicted hydrophobic interaction site. Binding was assessed by co-IP against the HA-tag on ConMem and immunoblotting against endogenous EMC4 in at least three independent replicates and normalized to V (mean ± SD, *P*-value < 0.05, ***P*-value < 0.001, two-tailed Student's *t* tests). **e** Cysteine crosslinking reveals spatial proximity of EMC1 and certain ConMem residues. Cysteines were introduced at position 963 of EMC1 and the indicated positions within the transmembrane domain of ConMem R13. Subsequently, crosslinking using the membrane-permeable maleimide crosslinker BMH was performed in cells and crosslinked species isolated by immunoprecipitation against HA (ConMem). EMC1:ConMem crosslinks could not be observed when co-transfecting ConMem Cys variants together with wildtype EMC1 or with empty vector, or when co-transfecting EMC V963C with ConMem R13 not containing a cysteine. Crosslinked species are highlighted by red arrows at the expected molecular weights, and the predicted molecular weights for EMC1 and EMC1 crosslinked with ConMem (EMC1xConMem) are indicated. **f** Mutations of the EMC1 chaperone site as in **c** also disrupts binding to natural clients identified in Fig. 1c. Binding was assessed by immunoprecipitation against FLAG (EMC1) and immunoblot against Strep (SIGMAR1, TMEM38B, BCAP31, SLC3A2) in at least three independent replicates and normalized to EMC1 wt (mean ± SD, *P*-value < 0.05, **P*-value < 0.01, ***P*-value < 0.001, ****P*-value < 0.0001, two-tailed Student's t tests). **g** Knockdown of EMC1 significantly destabilizes several natural clients at endogenous levels. Endogenous protein levels were assessed by immunoblots in seven independent replicates and normalized to a loading control (GAPDH or Hsc70) before comparison to a non-targeting siRNA control (mean ± SD, *P*-value < 0.05, **P*-value < 0.01, ***P*-value < 0.001, ******P*-value < 0.000001, two-tailed Student's *t* tests).

displayed a much smaller tilt in the membrane to begin with (Fig. 5a and Supplementary Fig. 7a). To test if such behavior was also observed for natural clients, we used TMD4 of TM4SF1 and TMD3 of SLC4A2, which we had found to strongly bind to the EMC (see Fig. 3c). In agreement with our observations of ConMem, binding to the EMC led to a more upright orientation in the membrane for TMD3 of SLC4A2, but not for TMD4 of TM4SF1 (Fig. 5b and Supplementary Fig. 7b). Of note, these natural clients showed a similar binding signature to EMC1 in MD simulations like our model ConMem R13 (Supplementary Fig. 7c, d). Together, this suggests that some EMC client TMDs may have a tendency to adopt tilted orientations in the lipid bilayer, e.g., due to snorkeling of polar residues. The EMC can then recognize TMDs with a tilted orientation, bind the TMD, and modulate the TMD orientation.

**Proper membrane protein assembly reduces EMC binding and allows transport to the cell surface**

Our data suggest that the EMC chaperones TMDs with challenging biochemical features, such as membrane-exposed polar residues, and that EMC engagement can affect TMD orientation in the membrane. To further understand this chaperoning function of the EMC in protein folding and assembly, we decided to study the role of the EMC in an intramembrane assembly process of two individual TMDs (Fig. 6a). To this end, we employed our ConMem system in an extended approach: We used the ConMem R13 variant, characterized by strong EMC binding, and co-expressed a second ConMem TMD variant in stoichiometric amounts using a P2A ribosomal skipping sequence (Fig. 6a, b and Supplementary Fig. 7e). When we co-expressed a second ConMem variant with a central Asp residue (D13 or D17), strong interaction with ConMem R13 was observed, accompanied by significantly reduced EMC binding (Fig. 6b and Supplementary Fig. 7e). For residues in the second ConMem that would not be expected to productively assemble with ConMem R13, either due to steric reasons (D6) or biochemical characteristics (Y9 or R13), no strong TMD assembly and no reduced EMC binding of ConMem R13 was observed (Fig. 6b and Supplementary Fig. 7e). Since some of the co-expressed constructs which did not reduce binding of the R13 variant to EMC also bind to the EMC with high affinity, these data also show that our assay reports on productive TMD interactions that reduce EMC binding and not on simple substrate competition for the EMC. Together, these data further support the idea that the EMC chaperones unassembled TMDs to allow for proper interaction in the membrane. Whether these

productive TMD interactions occur initially, precluding EMC binding, or if the individual TMDs are set free from the EMC upon pairing in the membrane cannot be determined based on this assay.

To assess the functional consequences of adding a stabilizing partner to an intrinsically unfavorable TMD, cell surface-exposed ConMem R13 was stained in flow cytometry assays, and the ratio to total expressed ConMem was calculated (Fig. 6c and Supplementary Fig. 7f). Both D13 and D17 significantly increased the proportion of ConMem R13 transported to the cell surface, whereas other variants that did not strongly assemble with R13 failed to promote transport to the cell surface. This interpretation was consistent with immunofluorescence microscopy experiments, where the localization of the different ConMem pairs agreed with our flow cytometry experiments (Supplementary Fig. 7g). Extending this data to a natural client, we investigated SLC3A2, an EMC1 binder identified in our crosslinking IP-MS experiment (Fig. 1c, Supplementary Fig. 2a, b). SLC3A2 can assemble with its cognate partner SLC7A5 into a heterodimeric amino acid transporter[35], which we confirmed by co-IP (Fig. 6d). In a subsequent second IP step, we isolated the fraction of EMC1-bound SLC3A2, which shows no SLC7A5 binding. When performing the co-IP in reverse order and first pulling down EMC1 and second SLC3A2, no SLC7A5 binding was detected in either IP step, demonstrating that SCL7A5 does not bind to the EMC itself, but also not to SLC3A2 while bound to the EMC. This finding indicates that SLC3A2 exists in at least two different states in cells: an immature state bound to the EMC1 holdase functionality and a mature state that can assemble with SLC7A5. Thus, EMC binding to problematic/unassembled transmembrane domains goes hand in hand with ER retention, while productive TMD assembly reduces EMC binding, facilitating the transport of membrane proteins to the cell surface.

## Discussion

The EMC was initially identified in a yeast screen for factors whose deletion causes ER stress[22]. Later on, it was characterized as a membrane insertase for low-hydrophobicity TA sequences and terminal TMDs of multipass membrane proteins[5–7]. Further potential functions of the EMC remained ill-defined on a molecular level, although possible chaperone functions for the EMC have been proposed in several studies[7,11,14,26,29]. Our work now provides clear evidence for such a chaperone function of the EMC and defines the biochemical characteristics and molecular mechanisms of EMC chaperone client engagement. This helps in understanding why the range of membrane

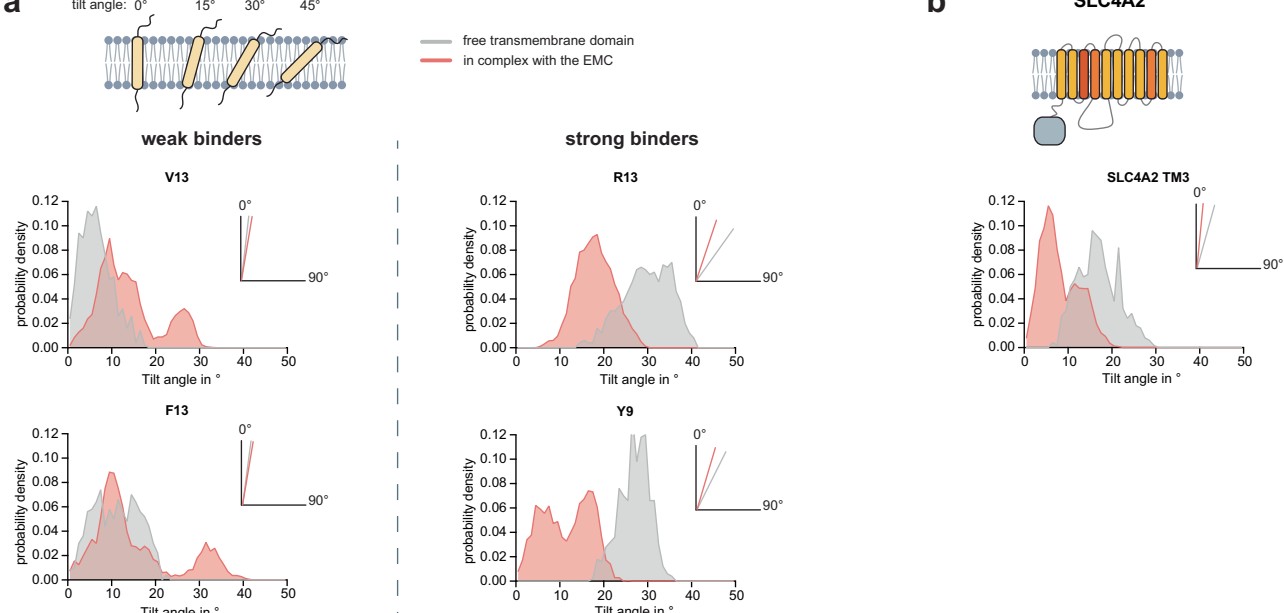

**Fig. 5 | EMC binding reorients transmembrane domains. a** Analysis of the TMD tilt towards membrane normal throughout molecular dynamics simulations (5 replicas for simulations in complex with EMC, and 1 replica for the free trans-membrane domain simulations, 100 ns simulation time per replica with tilt data collected every 200 ps) of selected ConMem variants in isolation (gray) and bound to the EMC1 chaperone site (red). Simulated TMDs with a high experimental EMC binding affinity (right) are tilted to higher angles in isolation, which is significantly reduced upon EMC binding. In contrast, the tilt angle of weak binders is not strongly affected by EMC binding (left). **b** Tilt angle analysis as in a, for TMD3 of SLC4A2, which was identified in Fig. 3c as a strong endogenous EMC binding transmembrane sequence. Please note that the topology of this protein is not well defined, and we show the one consistent with Uniprot annotation here, yet we cannot exclude a different orientation of the TMDs, which would affect our findings.

proteins whose biogenesis and stability depend on the EMC is larger than can be explained by its insertase functions alone[11–20].

Our data show that the lipid-filled cavity of the EMC, and in particular the single TMD of EMC1, is important for chaperone client binding. This is in agreement with a previous cryo-EM study on a complex of the EMC with an immature calcium channel[29]. Notably, our study now identifies critical residues in EMC1 that are important for client engagement. In previous work, they may have been difficult to define since the client engagement mode of EMC1 is complex. Through a combined in silico and experimental approach, we find that polar residues can induce a tilt in client TMDs and that EMC1 engages ER-oriented hydrophobic residues of such a TMD. This binding mode may also explain why proteins with low-hydrophobicity TMDs, often enriched in polar and charged amino acids, and often belonging to the class of ion channels, are particularly dependent on the EMC[11,12]. Also, our machine learning tool, ipred^EMC, predicts transporters to be particularly good binders of the EMC. This shows, together with our experimental validation, that useful predictions can be obtained with machine learning tools even when using limited datasets for training, like in this study. We would like to point out, however, that ipred^EMC needs further development and more extensive data for a more comprehensive capture of the EMC substrate range, as we mostly used simplified model proteins to train it. As a likely consequence, it shows a tendency for predicting TMDs with positively charged residues as binders, which were strong binders in our training set.

Together, our data support a model in which the EMC can bind challenging TMDs, e.g., those containing polar residues, but potentially also particularly long TMDs, and that binding to the EMC induces a more upright orientation of the bound TMD in the membrane. TMD binding will furthermore restrict the motility and orientation of interaction sites in such TMDs (e.g., unpaired charged/polar residues), potentially facilitating productive TMD interactions in the membrane (Fig. 7). Such a molecular mechanism of action may synergize with the insertase function of the EMC, ensuring that EMC-inserted TMDs are supported in assembling with the remainder of multipass membrane proteins, which are maintained in an assembly-competent conformation. This hypothesis is in agreement with recent findings on C-terminal TMD insertion by the EMC, which is facilitated by the binding of further TMDs of the same membrane protein to the EMC[7]. It is also in agreement with data on clients such as B1AR, which depend on the insertase function of the EMC but are also impaired by mutations at the now-identified chaperone site[11]. It is worth noting that we derived such a mechanism for single-pass TMDs, and that for multiple TMDs binding to the EMC simultaneously, it may differ. Indeed, although our work shows that binding of single TMDs is possible, many natural clients will likely engage the EMC chaperone site with multiple TMDs simultaneously or sequentially[29,31]. In agreement with this, our mass spectrometry interactomics show that single-pass and multipass membrane proteins bind to the EMC, but the latter may be more dependent on the EMC, as previous work[11] and our data show, which reveal that multipass membrane proteins generally tend to have higher predicted affinities for the EMC. Avidity effects for multipass membrane proteins when binding the EMC are likely, but different binding modes may also be possible, warranting further investigation in future studies. The reasons single-pass membrane proteins engage the EMC could vary. One possibility is that they need to assemble with other TMDs in the membrane. A role of the EMC in supporting protein assembly would be consistent with previous work on acetylcholine receptors[20] as well as calcium channels[29] and our data on SLC3A2. Another possibility is that EMC clients rely on its putative function as a chaperone recruitment platform, as our mass spectrometric inter-actome and previous work show it to be[11,21,36] similar to what has been observed for CNX[32]. It should also be noted that the site within the EMC we now identify as relevant for its intramembrane chaperone function was initially highlighted due to its property of binding lipids[26]. Thus, it could also play a role in the local modulation of the lipid bilayer

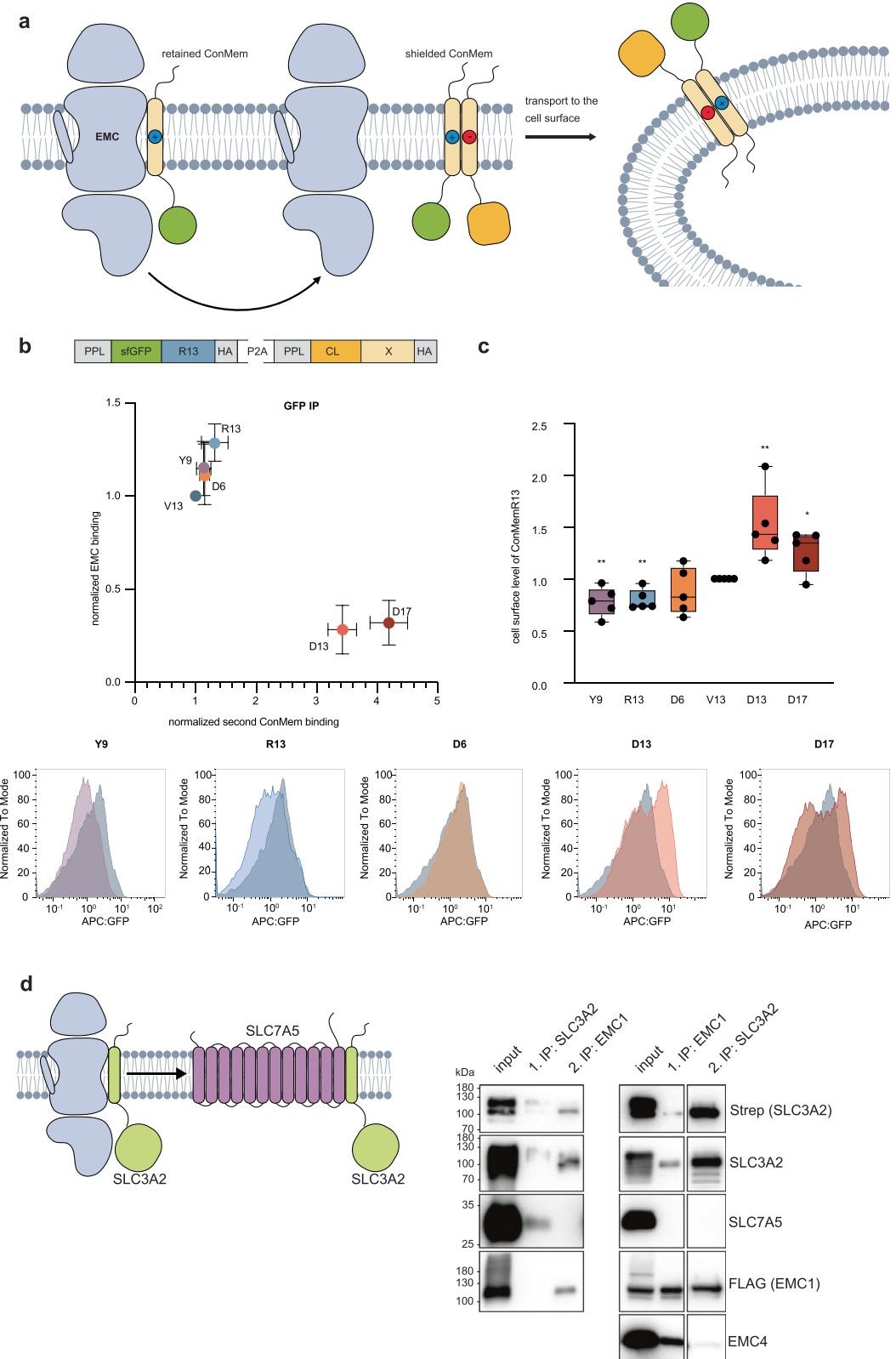

composition and, in this manner, affect membrane protein biogenesis. It is noteworthy in this regard that many enzymes involved in lipid biosynthesis and modification depend on the EMC[5,37,38].

One question that emerges from our study is whether dual occupancy of the EMC is possible, i.e. if it can process different insertase and chaperone clients at the same time, or if this is precluded by e.g. structural rearrangements upon binding one or the other (so

that it can e.g. synergistically integrate and chaperone the same TM protein as discussed above). Interestingly, mutations at the lipid-filled cavity of the EMC beneficially affect its insertase function[26], which may argue for competition between both engagement modes, be it on the level of EMC structural rearrangements or be it that certain chaperone clients also utilize EMC's insertase function (e.g. for their terminal TMDs), allowing the EMC to orchestrate multiple critical steps in

**Fig. 6 | Functional characterization of the EMC1 chaperone site as a holdase.** **a** Hypothesis underlying the experiments: When an EMC-interacting ConMem variant assembles with an oppositely charged, stabilizing counterpart inside the membrane, it should be released from the EMC and be transported to the cell surface if the EMC acts as a holdase chaperone. **b** Co-expression of stochiometric amounts of a second ConMem variant leads to assembly of some variants with the ConMem R13 (X-axis) and modulates EMC binding to the latter (Y-axis). Both ConMem variants were expressed from a shared promoter using a construct separated by a P2A sequence, one being a ConMem as described previously, and the other having its sfGFP moiety exchanged to an antibody light chain constant domain ($C_L$) (top). This allows assessment of interaction by IP-western experiments. Binding was assessed by immunoprecipitation against GFP in at least three independent replicates and normalized to ConMem V13 for both axes; error bars depict the SD. **c** Cell surface staining of ConMem R13 co-expressed with other variants indicates that strongly assembling ConMem pairs are efficiently trafficked to the cell surface. Surface transport was assessed by flow cytometry after cell surface staining with an APC-coupled anti-GFP antibody and calculating the APC:GFP ratio for each cell. This was assessed in five independent replicates, and the mean APC:GFP ratio was normalized to co-expressed $C_L$ V13 (box shows the interquartile range, whiskers reach from min to max values, horizontal line indicates the median, *P-value < 0.05, **P-value < 0.01, two-tailed Student's t tests). Representative histograms are shown below comparing the respective co-expressed $C_L$ ConMem variant to co-expressed $C_L$ ConMem V13 (in gray). **d** The validated EMC binder SLC3A2 (lime, validation in Supplementary Fig. 2a, b) assembles with its cognate partner SLC7A5 (purple) to form heterodimers. We assessed assembly of this heterodimer in sequential co-IPs: endogenous SLC7A5 can be co-IPed with SLC3A2 (1. IP left western blot, marked with a purple asterisk). However, no SLC7A5 is bound to the fraction of SLC3A2 bound to EMC1 (2. IP, left western blot). In the reverse IP order, SLC3A2 binds to EMC1 (1. IP, right western blot), and this complex is stable in another IP on SLC3A2 (2. IP, right western blot). SLC7A5 cannot be detected in either step, indicating that it does not bind to the EMC and furthermore also not to SLC3A2, which is bound to the EMC.

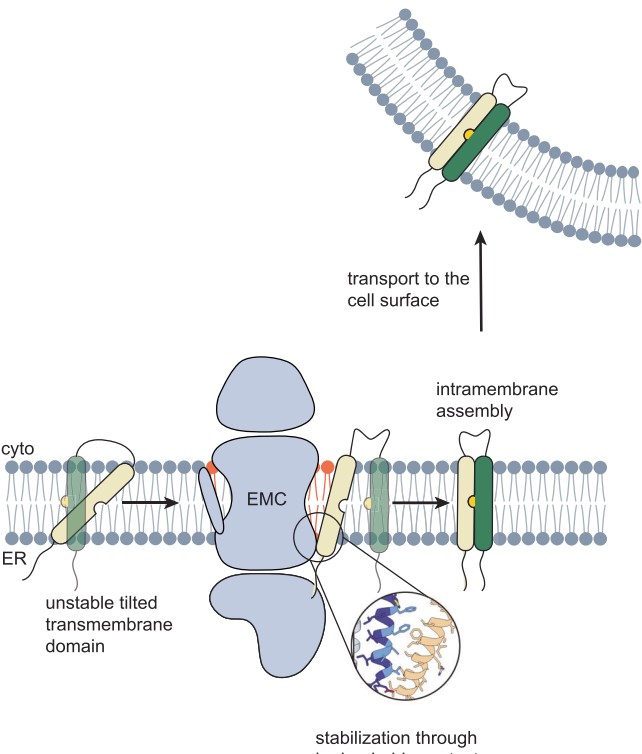

**Fig. 7 | The EMC acts as a chaperone for membrane proteins.** The chaperone site of the EMC recognizes transmembrane domains bearing suboptimal features, e.g. polar residues, which are often only marginally stable inside the lipid bilayer. Hydrophobic interactions with the EMC1 transmembrane domain stabilize and reorient these TMDs inside the membrane until assembly with a cognate partner occurs, which makes EMC binding no longer necessary and thus leads to release. The correctly folded and assembled membrane protein is no longer retained in the ER and can traffic to the cell surface. Note that the depicted processes are likely influenced and/or coupled to TMD insertion into the lipid bilayer.

membrane protein biogenesis with close temporal and spatial coordination. This would avoid misfolding, disassembly, and degradation of membrane proteins, processes that must be tightly controlled and regulated in any cell.

Taken together, from our study and previous work[8,36,37], the picture of the EMC as a multi-faceted molecular biogenesis and quality control platform for membrane proteins emerges. The EMC acts as a TMD insertase, an intramembrane chaperone, and additionally recruits many quality control factors to the membrane proteins it acts on. It will now be key to define how the different EMC functions are orchestrated on a molecular level, how the EMC synergizes with other machineries for membrane protein biogenesis[24], and how this impacts the fate of a protein: biogenesis versus ERAD, to maintain the fidelity of the membrane proteome.

## Methods

### DNA constructs
DNA was obtained from the following sources: ConMem wt[32], $C_L$ ConMem, and EMC1-myc from GeneArt Synthesis, Thermo Fisher; BCAP31, VMA21, TOR1AIP2, SIGMAR1, SLC3A2, TMX4, GPX8, and TMEM38B from Twist Bioscience. EMC1-FLAG was a kind gift of Bastian Bräuning[26]. pIRE4-BpaRS was a kind gift of Irene Coin[39]. All other constructs were generated and derived from these using standard molecular biology procedures and Gibson assembly and validated by Sanger sequencing.

### Cell culture and transient transfection
HEK293T cells (Sigma-Aldrich, 12022001) and COS-7 cells (African green monkey, Sigma-Aldrich 87021302) were maintained in Dulbecco's modified Eagle Medium (DMEM) high glucose (Sigma-Aldrich) supplemented with 10% (v/v) fetal bovine serum (Gibco) and 1% (v/v) antibiotic–antimycotic supplement (Gibco; final concentration 100 units/ml penicillin, 0.1 mg/ml streptomycin and 2.5 µg/ml Amphotericin B) at 37 °C with 5% $CO_2$. For transient transfection experiments, 600,000 cells were seeded per p35 well 24 h prior to transfection. Transient transfections were carried out using Metafectene Pro (Biontex) according to the manufacturer's instructions, using 4 µl Metafectene Pro per µg DNA. Protein expression was analyzed 24 h after transfection.

### siRNA-mediated knockdown
siRNA-mediated depletion was performed for 96 h using Lipofectamine RNAiMAX (Thermo Fisher) according to the manufacturer's instructions. In brief, cells were reverse-transfected with 60 pmol of siRNA (negative control 1, EMC1 s22953, both Thermo Fisher) per 400,000 cells and incubated for 48 h before a second round of siRNA treatment was performed under the same conditions.

### Photocrosslinking
24 h prior to transfection, 600,000 293T cells were seeded in poly-ᴅ-lysine-coated p35 plates. On the day of transfection, the medium was exchanged and supplemented with 25 mM HEPES, pH 7.5, and 250 µM H-p-Bz-Phe-OH (Bpa, Bachem), which was diluted from a 10 mM stock freshly dissolved in 0.5 M NaOH. Equal amounts of tRNA and tRNA synthetase encoding vector (pIRE4)[40] and the vector encoding for the gene of interest containing the amber codon were transiently transfected, and expression was analyzed after 24 h unless indicated otherwise. For photocrosslinking, cells were transferred on ice, briefly

washed with phosphate-buffered saline (PBS), and covered with 1 ml PBS per well before a handheld UV lamp was positioned 3 cm above the cells for 15 min. Cell lysis and immunoprecipitation were performed as described below using magnetic anti-myc beads (Chromotek), except that washing of the beads was carried out using 1X RIPA buffer (50 mM Tris pH 7.5, 150 mM NaCl, 1% Nonidet P-40, 0.5% DOC, 0.1% SDS).

## Cysteine crosslinking

Transiently transfected cells were washed once with ice-cold PBS, followed by the addition of 500 µM bismaleimidohexane (BMH, Thermo Fisher) in PBS per sample, diluted from a 20 mM stock of BMH in anhydrous DMSO. Crosslinking was performed at room temperature in the dark for 1 h, including mild agitation. The crosslinking reaction was quenched by washing the samples with PBS supplemented with 5 mM 2-mercaptoethanol for 15 min. Subsequently, the cells were washed twice with PBS, and cell lysis and immunoprecipitation were performed as described in the relevant sections.

## Lysine crosslinking

Cells were transferred onto ice, washed with ice-cold PBS before incubation with 1 mM dithiobissuccinimidylpropionate (DSP, Thermo Fisher) in PBS for 1 h on ice with occasional agitation. Crosslinking was quenched with 20 mM Tris, pH 7.5, in PBS for 15 min prior to cell lysis and immunoprecipitation as described below.

## PNGaseF digest

For deglycosylation experiments, cells were immediately transferred onto ice, washed twice with ice-cold PBS, and lysed in lysis buffer (50 mM Tris pH 7.5, 300 mM NaCl, 0.5% glyco-diosgenin (GDN101, Anatrace), 1x cOmplete EDTA-free protease inhibitor cocktail (Roche)). Insoluble material was removed via centrifugation, and 9 µl of the supernatant was denatured by the addition of 1 µl of 10x denaturation buffer (New England Biolabs) and from then on treated according to the manufacturer's instructions. The samples were then incubated at 37 °C for 1 h before they were supplemented with 5x Lämmli buffer with 10% (v/v) β-Mercaptoethanol to a final concentration of 1x and further incubation at 60 °C for 10 min.

## Immunoprecipitation

Cells were immediately transferred onto ice, washed twice with ice-cold PBS, and lysed in lysis buffer (50 mM Tris pH 7.5, 300 mM NaCl, 0.5% glyco-diosgenin (GDN101, Anatrace), 1x cOmplete EDTA-free protease inhibitor cocktail (Roche)). Insoluble material was removed via centrifugation, and the supernatant was rotated at 4 °C with 20 µl pre-coupled beads directed against epitope tags for 90 min. Beads used in this study: Pierce anti-HA magnetic beads (Thermo Fisher), Pierce anti-c-myc magnetic beads (Thermo Fisher), MagStrep Strep-Tactin XT beads (IBA), GFP-Trap magnetic agarose (Chromotek), FLAG M2 magnetic beads (Sigma Aldrich), Pierce Protein A magnetic beads (Thermo Fisher), StrepTactin 4Flow resin (IBA). A fraction of the input was kept for analytical purposes and supplemented with 5x Lämmli buffer with 10% (v/v) β-Mercaptoethanol to a final concentration of 1x and incubated at 60 °C for 10 min. Unbound material was washed off with wash buffer (50 mM Tris, pH 7.5, 300 mM NaCl, 0.01% GDN101). Bound proteins were eluted with 2x Lämmli buffer at 60 °C for 10 min.

## Sequential co-immunoprecipitation

The first pulldown was performed as described above for immuno-precipitation. Instead of a denaturing elution, a native elution was chosen using either 5 mM D-desthiobiotin (for anti-Strep beads) or 0.25 mg/ml FLAG peptide (Sigma Aldrich) (for anti-FLAG beads) in the wash buffer. Elution fractions were pooled and subjected to another round of immunoprecipitation against the second epitope tag. Bound proteins were then eluted with 2x Lämmli buffer at 60 °C for 10 min.

## SDS−PAGE and Western blot

Protein samples were separated on self-made 12% or pre-cast 4−20% gradient (Biorad #4561096) Tris-glycine SDS−PAGE gels and transferred onto a PVDF membrane (Biorad) at 30 V overnight. Membranes were blocked in 5% (w/v) skim milk powder in Tris-buffered saline with 0.05% (v/v) Tween-20 (TBS-T) and incubated with primary antibodies diluted in blocking buffer overnight at 4 °C. Membranes were washed and incubated for 1 h at room temperature with secondary HRP-coupled antibodies diluted 1:5000 in blocking buffer. Unbound secondary antibody was washed off prior to chemiluminescence detection using the Amersham ECL Prime Western blotting detection reagent (Cytivia) on a Fusion FX Edge Imager (Vilber Lourmat). Antibodies used in this study: EMC4 (Abcam, ab184544, 1:2000), EMC1 (Novus Biologicals, NBP2-59097 and NBP3-18427, 1:500), HA.11 (Biolegend, Poly9023, 1:1000), myc clone 4A6 upstate (Sigma-Aldrich, 05-724, 1:500), FLAG (Sigma-Aldrich, F7425, 1:1000), NWSHPQFEK 5A9F9 (Genscript, A01732, 1:1000), lambda-UNLB (Southern Biotech, SBA-1060-01, 1:500), TMEM38B (Proteintech, 19919-1-AP, 1:500), Sigmar1 (Proteintech, 15168-1-AP, 1:500), Slc3A2 (Proteintech, 15193-1-AP, 1:1000), BCAP31 (Proteintech, 11200-1-AP, 1:1000), Hsc70 (Santa Cruz Biotechnology, sc-7298, 1:500), EMC2 (Proteintech, 25443-1-AP, 1:500), ILVBL (Proteintech, 11220-1-AP, 1:500), PERK (Proteintech, 20582-1-AP, 1:500), GAPDH-HRP (Proteintech, HRP-60004, 1:4000), anti-rabbit IgG-HRP (Santa Cruz Biotechnology, sc-2357, 1:5000), m-IgGκ BP-HRP, sc-516102, 1:5000).

## Immunofluorescence

**Seeding and transfection.** 36 µl DMEM containing 3 µg DNA was mixed with 12 µl Metafectene Pro transfection reagent (Biontex) and incubated for 15 min at RT. 200 µl of COS-7 cell suspension with $3 \times 10^5$ cells/ml were added and mixed gently. 30 µl of the resulting suspension was added per inlet of a µ-Slide IV 0.4 (ibidi) and the µ-Slide was incubated for 3 h at 37 °C, 5% $CO_2$. Medium was replaced 3 h after seeding, after cell adhesion was visually verified. To this end, 60 µl complete DMEM was added per reservoir, and the µ-Slides were incubated for an additional 21 h at 37 °C, 5% $CO_2$.

**Staining.** For fixation, all liquid was removed from all reservoirs and channels, and 60 µl glyoxal fixation solution (20% EtOH, 7.825% glyoxal, 0.75% acetic acid; described in ref. [41]) was added. Samples were then incubated for 30 min on ice, followed by a further 30 min incubation at RT. The reaction was quenched by aspirating the fixation solution and adding 60 µl of 100 mM $NH_4Cl$ and subsequent incubation at RT for 20 min. Thereafter, samples were washed twice for 5 min with 100 µl 4 °C PBS. Then, 60 µl of blocking solution (3% BSA, 0.1% Triton X-100 in PBS) was added and samples were incubated for 5 min at RT following three further PBS washing steps. 30 µl of anti-Calnexin antibody (Santa Cruz Biotechnology, sc-23954) at 1:500 dilution in blocking solution was added and incubated for 2 h at RT, and subsequently washed three times for 5 min with 100 µl PBS. 30 µl of donkey polyclonal anti-mouse IgG antibody conjugated with TexasRed (Thermo Fisher, PA1-28626) at 1:300 dilution in blocking solution was then added and incubated for 1 h at RT in the dark. Alternatively to the anti-Calnexin and subsequent anti-mouse antibodies, cells were stained using an anti-PDI antibody directly coupled to Alexa647 (C-2, Santa Cruz Biotechnology, sc-74551) at a 1:100 dilution in blocking solution and then incubated for 1 h at RT in the dark. Excess antibody solution was then removed by washing thrice with 100 µl PBS. All liquid was removed, and 25 µl of DAPI solution (Sigma-Aldrich, 28718-90-3) (0.01% in PBS) was added to stain nuclei and incubated for 1 min. The samples were subsequently washed three times with PBS, the liquid was aspirated, and mounting medium (ibidi) was added to cover all inlets of the slides.

**Microscopy.** Imaging was performed on a DMi8 CS Bino inverted widefield fluorescence microscope (Leica) using a ×63 (NA = 1.4) oil

immersion objective or an attached STELLARIS 5 confocal microscope platform (Leica) using a ×63 (NA = 1.3) glycerol immersion objective. For the DMi8 microscope, the employed dichroic filters were chosen to image sfGFP (GFP channel; excitation/bandpass: 470/40 nm; emission/bandpass: 525/50 nm), TexasRed (TXR channel; excitation/bandpass: 560/40 nm; emission/bandpass: 630/75 nm), or DAPI (DAPI channel; excitation/bandpass: 350/50 nm; emission/bandpass: 460/50 nm). For the STELLARIS 5 confocal microscope, all dyes were imaged using the preinstalled dye settings (the GFP setting was chosen for sfGFP). LAS X (Leica) analysis software and Fiji ImageJ (NIH) were used for image analysis and processing. Adjustments of acquired images were restricted to homogenous changes in brightness and contrast over the whole image.

## Mass spectrometry

**Sample preparation.** After immunoprecipitation and washing steps, the magnetic beads with bound material were washed 1x with PBS, the buffer was removed and the beads were incubated for 20 min at 37 °C 100 μl with SDC buffer (1% sodium deoxycholate, SDC, 40 mM 2-chloroacetamide, CAA, 10 mM Tris(2-carboxyethyl)phosphine, TCEP, in 100 mM Tris, pH 8.0). Then, the samples were diluted with 100 μl of milliQ water, and the proteins were digested overnight at 37 °C by the addition of 0.5 μg trypsin (Promega). The supernatant was collected with the help of a magnetic rack and was acidified with trifluoroacetic acid (TFA) to a final concentration of 1%. Precipitated SDC was removed by centrifugation, and the peptide mixture was desalted via SCX StageTips. After elution, the samples were vacuum dried and dissolved in 15 μl of buffer A (0.1% formic acid). Samples (3 μl in volume) were loaded on Evotip Pure tips (Evosep).

**LC-MS/MS data acquisition.** Evotips were eluted onto a 15-cm column (PepSep C18 15 cm × 15 cm, 1.5 μm, Bruker Daltonics) via the Evosep One HPLC system. The column was heated to 50 °C, and peptides were separated using the 30 SPD method. Using the nanoelectrospray interface, eluting peptides were directly sprayed onto the timsTOF Pro mass spectrometer (Bruker Daltonics). Data acquisition on the timsTOF Pro was performed using timsControl. The mass spectrometer was operated in data-independent (DIA) PASEF mode. Analysis was performed in a mass scan range from 100 to 1700 $m/z$ and an ion mobility range from $1/K0 = 0.70$ to $1.30$ Vs cm$^{-2}$ using equal ion accumulation and ramp time in the dual TIMS analyzer of 100 ms each at a spectra rate of 9.52 Hz. DIA-PASEF scans were acquired a mass scan range from 350.2 to 1199.9 Da and an ion mobility range from $1/K0 = 0.70$ to $1.30$ Vs cm$^{-2}$. Collision energy was ramped linearly as a function of the mobility from 45 eV at $1/K0 = 1.30$ Vs cm$^{-2}$ to 27 eV at $1/K0 = 0.85$ Vs cm$^{-2}$. In complete, 42 diaPASEF windows were distributed to one TIMS scan each at switching Th precursor isolation windows, which led to an estimated cycle time of 2.21 s.

The ion mobility dimension was calibrated linearly using three ions from the Agilent ESI LC/MS tuning mix ($m/z$, $1/K0$: 622.0289, 0.9848 Vs cm$^{-2}$; 922.0097, 1.1895 Vs cm$^{-2}$; 1221.9906, 1.3820 Vs cm$^{-2}$).

## Data analysis

Raw data were processed using Spectronaut 18.0 (Biognosys) in directDIA+ (library-free) mode. Shortly, the peak list was searched against a predicted library of the human proteome database from Uniprot (downloaded in 2023). Cysteine carbamidomethylation was set as a static modification, and methionine oxidation and N-terminal acetylation as variable modifications. Proteins were quantified across samples using the label-free quantification (MaxLFQ) at the MS2 level.

The Perseus software package version 1.6.2.3 was used for further analysis. In brief, rows were filtered to remove single hits, values were log$_2$ transformed, and filtered to contain 3 valid values in at least one sample (3 replicates). Missing values were added from a normal distribution, and sample correlation and clustering were analyzed by principal component analysis (Supplementary Fig. 1d). For further processing, all 4 empty vector replicates were selected, as well as two replicates for each Bpa position, resulting in 6 samples for "EMC+Bpa ". These two groups were compared by a Student's $T$-test with the S0 parameter set to 2 and an FDR of 0.05. Candidates found to be significantly enriched in the EMC-Bpa samples were further analyzed based on their Uniprot and gene ontology (GO) annotations.

## Cell surface staining and flow cytometry

HEK293T cells were transiently transfected as described above with constructs encoding for sfGFP-ConMem R13 and a C$_L$-ConMem variant, separated by a P2A peptide skipping sequence. 24 h after transfection, cells were washed with PBS, gently dissociated from the growth plate, and incubated with an APC-coupled anti-GFP antibody (Biolegend, 338010) in the dark for 30 min at RT to stain surface-exposed sfGFP. Excess antibody was removed in two subsequent washing steps with PBS before the cells were analyzed on Attune NxT flow cytometer (ThermoFisher).

Flow cytometry data were analyzed in FlowJo 10 (BD Life Sciences). The gating strategy is visualized in Supplementary Fig. 7e by selecting healthy single cells according to behavior in forward and sideward scatter, excluding duplicate events, and gating for all GFP-positive cells (20–30% of all cells) as a marker for successful transfection. The APC:GFP ratio was derived for all GFP-positive cells from the respective signal heights.

## Development and use of the predictor of EMC:TMD binding affinity (ipred$^{EMC}$)

We identified 40 physicochemical and biochemical properties (Supplementary Table 3) of amino acid residues[33] that exhibited a significant correlation (absolute value of Pearson correlation coefficient more than 0.7 and $P$-value < 0.01) at the mutated positions of the TMD sequences in the ConMem dataset with the affinity of TMDs to the EMC. Feature values were scaled to the 0–1 range.

Feature values computed for each mutated position of a TMD were combined in a table along with their respective affinity values for training a predictor. The dataset was split into training (44 sequences) and test (15 sequences) subsets randomly. A multilayer perceptron (MLP) regressor was used for prediction, as implemented in scikit-learn (version 1.6.1). The model was trained using the default settings, with the exception of the maximum number of iterations, which was set to max_iter = 1000. The network architecture consisted of a single hidden layer with 100 neurons and employed the rectified linear unit (ReLU) activation function. Optimization was performed using the Adam solver with an initial learning rate of 0.001. L2 regularization was applied with a coefficient of 0.0001. The algorithm achieved a Pearson correlation coefficient of 0.9 between predicted and real values of affinity on the test set, showing that the predictor can rank TMDs based on their binding propensity to the EMC complex.

To globally analyze binders with this tool, all TMD sequences of proteins from the reference human proteome (Proteome ID in UniProt: UP000005640) were collected. The orientation of TMDs was defined according to Uniprot annotations or, if absent, predicted by Phobius[40]. TMD sequences were considered to start on the "exo" side of the membrane ('Extracellular', 'Lumenal' in UniProt annotations) as TMDs in the ConMem assay were inserted into the membrane in N$_{exo}$ orientation. Proteins predicted to be localized in other organelles were also analyzed. Parts of these proteins localized in the cytoplasm were considered to be in the "endo" environment, while other localizations −'Mitochondrial matrix', 'Peroxisomal matrix', 'Perinuclear space', 'Lumenal, vesicle', 'Intragranular', 'Lumenal, melanosome', 'Nuclear', 'Vesicular', and 'Mitochondrial intermembrane'−were considered to be "exo".

A second version of the predictor was trained and tested on 57 and 20 sequences, respectively, randomly selected from the extended

ConMem dataset (Supplementary Table 4). To overcome the challenge of different TMD lengths in the second version of the affinity predictor, all TMDs (including the entire human proteome and the ConMem dataset) were split into 9 subsequences. Only subsequences 2, 3, 5, 6, and 7 were used for further analysis as they correspond to the positions 6, 9, 13, 17, and 21 in the sequences of the ConMem set that contain amino acid substitutions. After testing different splits of TMD sequences, we found that using the predictor trained on 9 subsequences resulted in the highest correlation between predicted and measured values of a TMD's affinity to EMC. For each subsequence of each TMD feature, values were calculated as the average value over all residues in a subsequence. Feature values for segments of TMDs were aggregated into a table, along with their corresponding affinity values, to train the second iteration of the predictor using a multilayer perceptron regressor[42]. This model also achieved a high Pearson correlation coefficient of 0.9 between predicted and actual affinity values on a test set of 20 randomly selected ConMem TMD sequences. The finalized version of the predictor can be found in the Supplementary section.

## Simulation system model
The EMC protein complex was predicted as a multimer model with Alphafold 2[43] using the Alphafold database collection as of 06/22/2023 (Supplementary Fig. 1b). Sequences for the full-length EMC subunits 1–7, 9, and 10 were obtained from Uniprot (accessed on 07/10/2023, Supplementary Table 5)[44]. The Alphafold 2 model with the highest lDDT score (83.4) was used for subsequent MD simulations by embedding the protein into an ER membrane utilizing the CHARMM-GUI system preparation tools[45–47]. First, post-translational modifications were added to EMC1 in accordance with the Uniprot entry Q8N766. EMC1 Cysteines 227 ↔ 237 and 338 ↔ 368 were oxidized to disulfide bridges, and β-N-acetylgalactosamin glycosylations were added to EMC1-N370, EMC1-N818, EMC1-N913, and EMC10-N182. The TM region of EMC was predicted with the positioning of proteins in membranes (PPM3) method[48]. Afterwards, the protein complex was inserted with the CHARMM-GUI tool chain into an ER membrane containing 55% phosphatidylcholine (PC), 20% phosphatidylethanolamine (PE), 10% phosphatidylinositol, 7% cholesterol, 3% phosphatidylserine, 3% sphingomyeline, and 1% cardiolipin (Supplementary Table 6)[49,50]. The membrane composition was slightly asymmetric because the protein occupies a different volume in the upper and lower membrane leaflets. The membrane-protein system was then solvated in a $16.9 \times 16.9 \times 26.8$ nm$^3$ box. Subsequently, water molecules were replaced with sodium chloride ions to neutralize the system and adjust the ion concentration to 0.15 mol/l. Finally, the system was minimized and equilibrated for 200 ns in an isothermal–isobaric (NPT) ensemble.

The equilibrated EMC complex structure was removed from the equilibration system and aligned to the EMC1 subunit of the cryo-EM EMC structure in complex with the human Cav1.2 channel (pdb code: 8EOI[29]). Eight different simulation systems were built by aligning R13, V13, F13, Y9, P9, SLC4A2, or TM4SF1 to the human Cav1.2 channel TM helix closest to the EMC complex in the cryo-EM structure (amino acids 138–188). Two additional systems with F970A-F974A and V963K-S966K-S967K mutations in EMC1 and the TMD R13 were prepared. Each system was then inserted in an ER membrane, solvated, and neutralized in 0.15 mol/l sodium chloride as described above. Additionally, for each of the ten systems, five replicas were prepared that differed in the initial lipid starting positions, resulting in a total of 50 simulations. All systems were simulated for 100 ns in an NPT ensemble.

Another eight systems with R13, V13, F13, Y9, P9, SLC4A2, or TM4SF1 in the ER membrane are simulated as single replicas for 100 ns. All structural images were prepared using UCSF ChimeraX 1.6.1.

## Molecular dynamics simulation
The membrane protein systems were simulated with NAMD 3.0b3[51], representing proteins, lipids, and ions with the CHARMM36m force field[52–54], and water with the TIP3P model[55]. Periodic boundary conditions were applied during the simulation. Atomic movement was evaluated by a Verlet integrator every 2 fs with intermolecular interactions described by van der Waals and short-range electrostatic interactions for atom pairs within 1.2 nm, shifting the force to zero between 1 and 1.2 nm with a switching function. Long-range electrostatic interactions were calculated in a 0.12 nm spaced grid with the PME method[56]. Atom pairs within 1.4 nm were tracked in interaction pair lists that are generated every 200 fs. All bonds to hydrogen atoms were constrained with the SHAKE algorithm[57], except bonds in water molecules, which are constrained with the SETTLE algorithm[58]. A Langevin temperature bath[59] with a damping coefficient of 1 ps$^{-1}$ controls the temperature to 310 K, while a Nosé–Hoover Langevin piston[60] with a 200 fs oscillation period and 100 fs damping time scale adjusts the pressure semiisotropically to 1 atm.

## Contact number analysis
Interactions between TMD and EMC were evaluated using contact numbers $C_{ij}$, which are calculated based on the minimum periodic distances $d_{kl}$ between heavy atoms $k$ and $l$.

$$C_{ij} = \left\langle \sum_k \sum_l \frac{1}{1 + e^{5(d_{kl}-4)}} \right\rangle$$

Thereby, heavy atoms $k$ belong to residue $i$ in TMD and heavy atoms $l$ to residue $j$ in EMC. $\langle \rangle$ denotes the time average over the trajectory frames. Pair distances below 0.4 nm have contact numbers between 1 and 0.5, while larger distances approach zero.

## Orientational probability distributions
The rotational freedom of TMD in the ER membrane with respect to the membrane normal was analyzed using principal axis decomposition of all heavy atoms in the helical region, assuming that the helix is relatively stiff and does not bend significantly during the simulation.

## Quantification and statistical analysis
Densitometric quantification of western blots was performed either with ImageJ 1.54 Fiji or Bio-1D (Vilber Lourmat) version 15.08 d using standard settings, and background correction was performed using a rolling-ball algorithm with ball size recommended by the software. Unless indicated otherwise, statistical analyses were performed using GraphPad Prism version 10. Statistical analysis of protein levels in co-immunoprecipitation experiments was performed using an unpaired Student's t-test.

## Reporting summary
Further information on research design is available in the Nature Portfolio Reporting Summary linked to this article.

# Data availability
Further information and requests for resources and reagents should be directed to and will be fulfilled by the lead contact Matthias J. Feige (matthias.feige@tum.de). All primary data of this study will be shared upon request from the lead contact. Mass spectrometry data are available via ProteomeXchange with identifier PXD059792. Initial and final configurations from our MD simulations have been deposited on Zenodo (https://zenodo.org/records/15688608). Any additional information required to reanalyze the data reported in this work paper is available from the lead contact upon request. Source data are provided with this paper.

## Code availability

The code for the machine learning algorithm has been uploaded to Github (https://github.com/MarinaParr/ipredEMC).

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

## Acknowledgements

We are grateful to Johanna Pritzl and Felix Johannsen, TUM, for support with molecular biology procedures, to members of the lab for critical comments on the manuscript, to Bastian Bräuning, MPI of Biochemistry, for critical feedback on the manuscript and to Brenda Schulman, MPI of Biochemistry, for constructive feedback during the entire course of the study. We thank Irene Coin, Leipzig University, for sharing the pIRE4-BpaRS plasmid and for advising on its use. We thank the MPI of Biochemistry Mass Spectrometry Facility (RRID:SCR_025745). This project was funded by the DFG (FE 1581/5-1, to M.J.F.) and the European Union (ERC, DeCoDe, 101088970, to M.J.F.). Views and opinions expressed are, however, those of the author(s) only and do not necessarily reflect those of the European Union or the European Research Council. Neither the European Union nor the granting authority can be held responsible for them. C.J.K. gratefully acknowledges Ph.D. fellowships from the Boehringer Ingelheim Fonds and the Studienstiftung des deutschen Volkes.

## Author contributions

C.J.K. and K.M.B. performed all experiments in the study, MS experiments were performed together with B.S., M.K. performed MD simulations, M.P. performed bioinformatic analyses. M.Z. supervised MD simulations, D.F. supervised bioinformatic analyses, and M.J.F. supervised the experimental part of the study. The study was conceived by M.J.F. together with C.J.K. and K.M.B. The paper was written by C.J.K., K.M.B., and M.J.F. and edited by all authors.

## Funding

## Competing interests

The authors declare no competing interests.
