## [Transparent Peer Review file · Nature Communications]

The EMC acts as a chaperone for membrane proteins

Corresponding Author: Professor Matthias Feige

Version 0:

Reviewer comments:

Reviewer #1

(Remarks to the Author)

The ER membrane protein complex (EMC) has been previously shown to mediate the insertion of both N-terminal and C-terminal transmembrane domains (TMDs). Additionally, earlier studies have suggested that EMC functions as a membrane chaperone to promote membrane protein assembly, though the underlying mechanisms remain poorly understood. In this manuscript, Klose et al. employ photocrosslinking and mass spectrometry approaches to identify novel membrane protein clients that interact with EMC. Using a reductionist approach, the authors extensively characterize the biophysical properties of TMDs that determine their interaction with EMC. They propose that EMC1 preferentially interacts with charged TMDs, but unexpectedly, this interaction does not occur through direct pairing with the charged residues. Instead, the authors suggest that EMC1 engages these TMDs primarily through hydrophobic interactions. Overall, the presented data and model are intriguing. However, the evidence provided does not sufficiently support the proposed mechanism due to potential indirect effects arising from overexpression of membrane proteins and uncertainties regarding their orientation and localization in the ER membrane.

Major Concerns

1. The authors demonstrate that overexpressed membrane protein clients interact with EMC1 and EMC4 (e.g., Fig. 2a). However, overexpression of membrane proteins is known to result in non-specific interactions with other membrane proteins. To exclude this possibility, the authors should probe for endogenous Calnexin as a negative control.
2. The authors assume that the selected TMDs (e.g., Fig. 2c) insert into the ER membrane in the correct orientation. However, this should be experimentally validated. A common approach is to introduce a C-terminal glycosylation tag and assess whether these TMDs are properly inserted rather than completely translocated into the ER lumen. This validation is crucial, as many TMDs, when tested individually, can be entirely translocated into the lumen instead of inserted into the membrane. Additionally, many of the western blots show multiple bands for the substrates, but the authors do not explain these bands, complicating the interpretation of the data.
3. The authors report that TMD2 of BCAP31 exhibits strong binding to EMC (Fig. 1d), despite being highly hydrophobic and lacking charged residues, unlike TMD1 and TMD2, which contain charged residues. This finding contradicts the authors' model that EMC preferentially interacts with charged residues on TMDs. Instead, it suggests that EMC may non-specifically interact with TMD2 due to its high expression and stability in the ER membrane, whereas TMD1 and TMD2w may be less stable and thus engage with membrane quality control machinery. The authors should address this discrepancy.
4. The authors use a neural network (Fig. 3b) to predict strong EMC-interacting TMDs containing four charged residues and subsequently validate these interactions experimentally (e.g., Fig. 4c). However, TMDs with four charged residues are highly likely to be fully translocated into the ER lumen rather than inserted into the membrane. The authors attempt to address this concern by adding a glycosylation tag at the C-terminus, demonstrating that these TMDs remain membrane-inserted. However, a proper positive control for this assay is missing. The authors should construct a control where the TMD is replaced with a completely hydrophilic sequence and show that it can be completely translocated and receive N-glycosylation modifications. This is important because N-glycosylation efficiency can be sequence-dependent and may not always serve as a reliable indicator of translocation.
5. The authors should clarify whether the photocrosslinking between the strong ConMem variant (R13) and EMC1 occurs within the ER membrane or the ER lumen. There is a possibility that R13 is fully translocated into the lumen and interacts with the luminal domain of EMC1 rather than its membrane-embedded region. The authors should rule out this possibility.
6. The manuscript would be significantly strengthened if the authors demonstrated the assembly of the identified membrane protein clients rather than focusing solely on engineered ConMem TMDs (Fig. 6).

Minor Concerns

1. The authors should briefly describe their experimental approaches in the figure legends rather than discussing the results. For instance, the term “binding experiments” is vague—does this refer to immunoprecipitations following crosslinking or to experiments conducted without crosslinking?

(Remarks on code availability)

Reviewer #2

(Remarks to the Author)

The manuscript by Klose et al., provides important insights into the chaperone activity of the EMC, which is currently less well-characterised at the molecular level than its insertase activity. The authors use an impressive combination of site-specific crosslinking, mass spectrometry, co-immunoprecipitation (co-IP) and molecular dynamics (MD) simulations to identify transmembrane domains (TMDs) within artificial and natural membrane protein substrates that preferentially bind the EMC. The biochemical data are used to develop a machine-learning approach which accurately predicts EMC substrate binding within the human proteome. In line with evidence from existing literature, the authors show that chaperone substrate binding occurs via the EMC1 TMD within the lipid-filled cavity and propose a molecular model for the interaction. Overall, the manuscript is thorough, well-presented and of interest to a broad readership, therefore is suitable for publication given the following can be addressed/answered:

1. It would help to clarify in the legend for Figure 1c how many replicates were used for the enrichment analysis. The results were averaged over the three cross-linking positions for robustness; given the authors later show that the EMC1 TMD interface is the most important for the chaperone-substrate interaction, it would also be interesting to know whether there were significant differences between the interactomes from the three positions. Was CaV1.2 enriched?

2. Given that the same construct design is used for substrate co-IPs throughout the paper, why is the mode of misintegration in some experiments assumed to be insertion in the incorrect topology (Extended Data Figures 2c and 4d) and in other experiments complete translocation into the ER (Extended Data Figure 4b)? If misintegration of the model protein is by full ER translocation (which with a signal peptide and GFP at the N-terminus seems more likely), the PNGase F assay in Extended Data Figures 2c and 4d would be expected to look the same for correct and incorrect integration. Extension of the fluorescence microscopy shown in Extended Data Figure 2b would provide further support for correct integration of the constructs in Extended Data Figures 4b and 4d.

3. The co-IP experiment in Figure 3c would be more compelling if predicted weaker binding TMDs from CD9, TM4SF1 and/or SLC4A2 were included as a comparison.

4. Is the binding interface with the EMC1 TMD similar in the MD simulations with SLC4A2 TMD3 and TM4SF1 TMD4 to that shown in Figure 4b? It would be informative to show molecular snapshots of the MD simulations for these natural substrates and the strong binders in Figure 5 as further evidence for the designated chaperone interaction site.

5. Line 486: the authors state that ConMem mutants D6, Y9 and R13 lead to “no TMD assembly”, but Extended Data Figure 7c shows that the CL tagged variants are still pulled-down with these variants considerably, suggesting the statement is too strong. The schematic in Figure 4b shows both GFP and CL variants are HA-tagged, therefore it should be clarified what the bands in the HA western blot in Extended Data Figure 7c represent. Do the bands around the 55 kDa marker in the CL western blot correspond to ConMem variant fusions formed due to no peptide skipping at the P2A sequence? If so, the authors should show that this band is present for all variants (e.g. by increased exposure of the input) and not just for the D13 and D17 variants where different EMC binding/cell surface exposure is seen.

6. Line 499: again “other variants that did not assemble with R13 failed to promote transport to the cell surface” seems a strong statement given transport occurs to a lesser extent. The authors conclude that differential binding of ConMem variants to the EMC as assessed by in vitro co-IP (Figure 6b) is the reason for their differential cell-surface exposure (Figure 6c), however this correlation is rather indirect given a multitude of factors are likely at play in the trafficking of these constructs in vivo. To make the statement, “EMC binding to problematic transmembrane domains goes hand in hand with ER retention”, the authors should show supporting fluorescence microscopy for the strains in Figure 6.

(Remarks on code availability)

Reviewer #3

(Remarks to the Author)

The study investigates the ER membrane protein complex (EMC) as a chaperone for labile transmembrane domains (TMDs). The authors focused on the lipid-filled cavity of EMC, and its binding with various transmembrane proteins. Using interactomics, systematic mutagenesis, molecular dynamics simulations, and a machine learning tool (ipredEMC), the authors demonstrate that EMC binds TMDs via hydrophobic interactions, rectifies their orientation in the membrane, and facilitates assembly. The work is valuable in understanding EMC's dual roles. The integration of various techniques

demonstrates a strength leading to the discovery. The work in general is of good quality. I have some comments for the authors' further consideration.

1. It was found that three positions within EMC1 showed most crosslinks: at the brace/crossbar (R513), at the the luminal side of the single EMC1 TMD (D961), and at its cytoplasmic side (R981). Further discussion can be provided on this finding. Why the three positions are specific.
2. The authors employed ConMem to investigate the interaction between EMC and TMDs. It should better explain the reason of choosing the variants sites shown in Figure 2c to study the interaction.
3. The machine learning model should be explained better in detail. As the datasets for model training is not large, there can be problems like overfitting. Can the machine learning prediction be tested with the proteomic results shown in Figure 1c, to explain the significantly enriched transmembrane proteins?
4. The authors note that ~50% of top ipredEMC hits are transporters (Fig. 3e) but do not discuss why this class is particularly reliant on EMC. Are their TMDs enriched in polar residues or other instability features?
5. In Fig. 4e, the absence of crosslinks in wild-type EMC1 controls is not quantified. A negative control (e.g., non-binding TMD) would strengthen this experiment.
6. The study posits that EMC binding "rectifies" TMD orientation (Fig. 5), but the functional consequence of this reorientation (e.g., enhanced folding or assembly) is not directly tested.
7. The link between EMC's insertase and chaperone functions remains speculative. Competition experiments (e.g., co-expressing insertase and chaperone clients) could address whether these roles are mutually exclusive or synergistic.
8. As the work is rather complicated with the combination of various techniques, a summary of the main finding at the end of the discussion section is valuable.

(Remarks on code availability)

The codes are clear.

Version 1:

Reviewer comments:

Reviewer #1

(Remarks to the Author)

The authors have addressed my concerns, and I have no concerns.

(Remarks on code availability)

Reviewer #2

(Remarks to the Author)

The revised manuscript by Klose et al., contains several additional control experiments and further analyses in response to the reviewer's comments. As a result, the data are compelling, significantly advance our understanding of the EMC and are ready for publication. I have a few minor comments that would improve clarity/interpretability for the reader:

1. I would explicitly name the constructs that are misintegrated and excluded from further analyses in the legends of Supplementary Fig. 2c and 4e.
2. For Supplementary Fig. 2c and 4e, the cytosolic glycosylation site would also give a shift upon PNGase treatment if the construct were misintegrated in the wrong topology, therefore the top panels are sufficient to reveal the misintegrated constructs. For simplification, I would suggest to remove the now-redundant bottom panels showing the results for the luminal glycosylation site. The cartoon showing misintegration in the wrong topology could be included as an alternative scenario in the top panel.
3. On page 12, I would suggest to append a statement similar to "although the extent of binding does not directly correlate with the predicted affinities" to the sentence "we found that TMDs which ipredEMC predicted to bind to the EMC indeed did", as Fig. 3c is still designed to show the affinity predictions for the single TMDs of the multipass proteins.
4. On page 15, I would (perhaps in brackets) explicitly name the residues referred to by "the positions within EMC1 which were predicted to form the most contacts with residues of ConMem", "residues previously proposed to be important for chaperone client engagement by the EMC" and "we introduced mutations at the EMC1 brace" as it would really help with interpreting the data in Supplementary Fig. 5.
5. On page 16, I would change "while more distal mutants did less so" to "while more distal mutants did not" as Fig. 4d does not show a significant change in EMC binding for the T11 and T15 variants.

(Remarks on code availability)

Reviewer #3

(Remarks to the Author)

My previous concerns have been addressed. I have no further questions. It would be better if the authors provide a highlighted version of the revised manuscript to illustrate the changes.

(Remarks on code availability)

The codes are clear.

First and foremost, we would like to thank all three reviewers for the time they have invested in reviewing our manuscript. Addressing their constructive comments has helped us to further improve our work. In the following, please find a detailed reply to all the points that have been raised during the review process:

Reviewer #1 (Remarks to the Author):

The ER membrane protein complex (EMC) has been previously shown to mediate the insertion of both N-terminal and C-terminal transmembrane domains (TMDs). Additionally, earlier studies have suggested that EMC functions as a membrane chaperone to promote membrane protein assembly, though the underlying mechanisms remain poorly understood. In this manuscript, Klose et al. employ photocrosslinking and mass spectrometry approaches to identify novel membrane protein clients that interact with EMC. Using a reductionist approach, the authors extensively characterize the biophysical properties of TMDs that determine their interaction with EMC. They propose that EMC1 preferentially interacts with charged TMDs, but unexpectedly, this interaction does not occur through direct pairing with the charged residues. Instead, the authors suggest that EMC1 engages these TMDs primarily through hydrophobic interactions. Overall, the presented data and model are intriguing. However, the evidence provided does not sufficiently support the proposed mechanism due to potential indirect effects arising from overexpression of membrane proteins and uncertainties regarding their orientation and localization in the ER membrane.

We thank Reviewer #1 for the positive evaluation of our manuscript and thoughtful feedback. In response to the reviewer's concerns, we have included additional data to address potential issues arising from protein overexpression and extensively validated the topology of model proteins used in this study. Furthermore, we have made textual adaptations to improve clarity and the discussion of our findings.

Major Concerns

1. The authors demonstrate that overexpressed membrane protein clients interact with EMC1 and EMC4 (e.g., Fig. 2a). However, overexpression of membrane proteins is known to result in non-specific interactions with other membrane proteins. To exclude this possibility, the authors should probe for endogenous Calnexin as a negative control.

We agree with Reviewer #1 that overexpression of membrane proteins often leads to unspecific interactions. To address this, we validated the interaction of the EMC with a selected number of clients at endogenous levels in an endogenous IP against EMC4 followed by immunoblotting for endogenous levels of the respective clients. This data is now included in the new Supplementary Figure 2b.

Our own unpublished data, as well as that of others, have shown that Calnexin interacts with the EMC at endogenous levels. Therefore, instead of Calnexin, we have selected ILVBL as a dis-enriched hit from our mass spectrometry and PERK, another ER membrane resident protein, as negative controls for EMC interaction (now also included in Supplementary Figure 2b).

2. The authors assume that the selected TMDs (e.g., Fig. 2c) insert into the ER membrane in the correct orientation. However, this should be experimentally validated. A common approach is to introduce a C-terminal glycosylation tag and assess whether these TMDs are properly inserted rather than completely translocated into the ER lumen. This validation is crucial, as many TMDs, when tested individually, can be entirely translocated into the lumen instead of inserted into the membrane. Additionally, many of the western blots show multiple bands for the substrates, but the authors do not explain these bands, complicating the interpretation of the data.

These are indeed valid points, as also our own work in the TCR alpha chain has shown in the past that membrane proteins can fully translocate into the ER lumen. We thus comprehensively analyzed our model systems and in this respect i) added a table of the predicted ΔG_{app} values for membrane integration (according to Hessa et al, Nature, 2007) (new Supplementary Table S2) and ii) performed glycosylation assays for a C-terminal site for our models, especially those constructs with predicted ΔG_{app} values close to 0 or above 0. Indeed, we found a few constructs not to be integrated quantitatively by these approaches. We thus thank the reviewer for this critical suggestion. Incompletely integrated constructs have now been entirely removed from any further analyses in this study (BCAP31 TMD1 and TMD3 in Supplementary Figure 2d and CD9 TMD2 and TM4SF1 TMD2 in Supplementary Figure 4f).

The presence of multiple bands for the employed ConMem constructs can be explained by refolding/incomplete unfolding of the sfGFP moiety. This is analyzed in greater detail in Bloemeke et al., EMBOJ, 2022. In addition, for some membrane proteins lysis and denaturation impact the band pattern (by partial aggregation, see below in the reply 5 to reviewer 2).

3. The authors report that TMD2 of BCAP31 exhibits strong binding to EMC (Fig. 1d), despite being highly hydrophobic and lacking charged residues, unlike TMD1 and TMD2, which contain charged residues. This finding contradicts the authors' model that EMC preferentially interacts with charged residues on TMDs. Instead, it suggests that EMC may non-specifically interact with TMD2 due to its high expression and stability in the ER membrane, whereas TMD1 and TMD2 may be less stable and thus engage with membrane quality control machinery. The authors should address this discrepancy.

We thank the reviewer for raising this important point. We have re-investigated the individual TMDs of BCAP31, also in the light of additional topology assays conducted, and found indeed TMDs 1 and 3 to not fully integrate into the ER membrane in isolation. This likely explains the reduced interaction of the EMC with these two. Due to their incomplete integration, we have decided to exclude these constructs from our manuscript.

4. The authors use a neural network (Fig. 3b) to predict strong EMC-interacting TMDs containing four charged residues and subsequently validate these interactions experimentally (e.g., Fig. 4c). However, TMDs with four charged residues are highly likely to be fully translocated into the ER lumen rather than inserted into the membrane. The authors attempt to address this concern by adding a glycosylation tag at the C-terminus, demonstrating that these TMDs remain membrane-inserted. However, a proper positive control for this assay is missing. The authors should construct a control where the TMD is replaced with a completely hydrophilic sequence and show that it can be completely translocated and receive N-glycosylation modifications. This is important because N-glycosylation efficiency can be sequence-dependent and may not always serve as a reliable indicator of translocation.

We agree with the reviewer that a proper positive control is essential to assess N-glycosylation efficiency. We have designed a control based on our poly-L TMD construct by adding several charged residues to achieve a high predicted ΔG_{app} for membrane insertion according to ΔG_{pred} was achieved. As expected, this construct was not integrated into the membrane, and importantly, was C-terminally glycosylated. In contrast, all the binders we used to validate our algorithm were properly integrated (see Supplementary Table S2 and Supplementary Figure 4d).

5. The authors should clarify whether the photocrosslinking between the strong ConMem variant (R13) and EMC1 occurs within the ER membrane or the ER lumen. There is a possibility that R13 is fully translocated into the lumen and interacts with the luminal domain of EMC1 rather than its membrane-embedded region. The authors should rule out this possibility.

We thank the reviewer for raising this important point. Indeed, using the Bpa approach we cannot determine the site of crosslinking to EMC1. However, we have validated the interaction interface using cysteine crosslinking (Fig. 4e and Supplementary Fig 6b). Here, we clearly see Cys crosslinks between EMC1 and ConMem R13 inside the lipid bilayer. Furthermore, our now extended glycosylation assays confirm the correct membrane integration of ConMem R13 and two of the R13-based cysteine variants (Supplementary Figure 6a), together supporting the notion of intramembrane crosslinking

6. The manuscript would be significantly strengthened if the authors demonstrated the assembly of the identified membrane protein clients rather than focusing solely on engineered ConMem TMDs (Fig. 6).

We appreciate this excellent suggestion, as this data would be further compelling evidence to support our chaperone hypothesis. We thus decided to investigate SLC3A2 which we have identified as EMC1 binder in our crosslinking IP-MS experiment (Fig. 1). SLC3A2 is known to assemble into heterodimers with a cognate partner, SLC7A5; the heterodimers are localized to the cell surface. We reasoned that this assembled heterodimer would bind less to the EMC.

To test this hypothesis, we performed sequential co-IP experiments now included in a new Fig. 6d. In a first step we isolated SLC3A2 from 293T cells, which we found to interact with EMC1 as well as endogenous levels of SLC7A5. In a subsequent step we isolated the fraction of EMC1-bound SLC3A2 which shows no SLC7A5 binding. When performing the co-IP in reverse order and first pulling down EMC1 and then SLC3A2 we could not detect any SLC7A5 binding, indicating that a) SLC7A5 binds to SLC3A2, but not when the latter is bound to the EMC and b) SLC7A5 does not bind to the EMC. This finding indicates that at least two states of SLC3A2 exist: an immature state, bound to the EMC holdase functionality and a mature state that can assemble with SLC7A5. We have added this new finding to Figure 6 as panel d. This finding is also consistent with report in the literature by Chen et al., Nature, 2023, which demonstrates that when the client CaV1.2 subunit is bound to the EMC1 holdase function, binding of the respective beta subunit is structurally not possible.

Minor Concerns

1. The authors should briefly describe their experimental approaches in the figure legends rather than discussing the results. For instance, the term “binding experiments” is vague—does this refer to immunoprecipitations following crosslinking or to experiments conducted without crosslinking?

To address this comment, we have revised the figure legends accordingly to describe the experimental approaches more clearly. If not directly stated, all immunoprecipitations were carried out without crosslinking.

Reviewer #2 (Remarks to the Author):

The manuscript by Klose et al., provides important insights into the chaperone activity of the EMC, which is currently less well-characterised at the molecular level than its insertase activity. The authors use an impressive combination of site-specific crosslinking, mass spectrometry, co-immunoprecipitation (co-IP) and molecular dynamics (MD) simulations to identify transmembrane domains (TMDs) within artificial and natural membrane protein substrates that preferentially bind the EMC. The biochemical data are used to develop a machine-learning approach which accurately predicts EMC substrate binding within the human proteome. In line with evidence from existing literature, the authors show that chaperone substrate binding occurs via the EMC1 TMD within the lipid-filled cavity and propose a molecular model for the interaction. Overall, the manuscript is thorough, well-presented and of interest to a broad readership, therefore is suitable for publication given the following can be addressed/answered:

We thank Reviewer #2 for the encouraging feedback and positive evaluation of our manuscript. To address the reviewer's questions, in addition to text edits, we have added more detailed analyses of the molecular dynamics simulations and mass spectrometry data and provide a more thorough rationale for the data processing strategy chosen. Moreover, we have included additional experimental evidence to address concerns about subcellular localization, potential misintegration or full translocation of model proteins, and have analyzed the binding of additional suggested sequences.

1. *It would help to clarify in the legend for Figure 1c how many replicates were used for the enrichment analysis. The results were averaged over the three cross-linking positions for robustness; given the authors later show that the EMC1 TMD interface is the most important for the chaperone-substrate interaction, it would also be interesting to know whether there were significant differences between the interactomes from the three positions. Was CaV1.2 enriched?*

We prepared three replicates for each position of Bpa incorporation as well as four replicates for the empty vector control. During data evaluation and processing, the individual replicates were analyzed by principal component analysis (PCA) and the least clustering replicate for each position was excluded, resulting in a total of 6 replicates. We have now added the PCA plot to new Supplementary Figure 1d. The PCA plot indicates differences between the individual positions, yet an overall similarity.

Figure 1: Venn Diagram of significantly enriched interactors of EMC1 variants bearing the unnatural photocrosslinking amino acid Bpa at positions R513, D961 or R981, respectively, compared to empty vector. This data was analyzed from all measured replicates.

To specifically address the reviewer's question about the differences between the crosslinking positions, we are including a Venn diagram overview of significantly enriched interactors at the individual positions before exclusion of any replicates. We did not detect CaV1.2 in our measurements and therefore cannot comment on enrichment.

2. Given that the same construct design is used for substrate co-IPs throughout the paper, why is the mode of misintegration in some experiments assumed to be insertion in the incorrect topology (Extended Data Figures 2c and 4d) and in other experiments complete translocation into the ER (Extended Data Figure 4b)? If misintegration of the model protein is by full ER translocation (which with a signal peptide and GFP at the N-terminus seems more likely), the PNGase F assay in Extended Data Figures 2c and 4d would be expected to look the same for correct and incorrect integration. Extension of the fluorescence microscopy shown in Extended Data Figure 2b would provide further support for correct integration of the constructs in Extended Data Figures 4b and 4d.

We apologize for any confusion arising from the way of presenting our data and agree with the reviewer that the assessment of membrane integration is crucial. We thus, as suggested, now include further microscopy studies in Extended Figure 2, which agree with our interpretations.

Additionally, we have included further predictions (ΔG_{app} for membrane integration, Hessa et al., Nature, 2007) and several more experiments to assess whether our constructs are correctly integrated or fully ER-translocated, including a series of new positive controls (see Extended Table S2 and Supplementary Figures 2d, 3d and 4d).

3. The co-IP experiment in Figure 3c would be more compelling if predicted weaker binding TMDs from CD9, TM4SF1 and/or SLC4A2 were included as a comparison.

We thank the reviewer for this good suggestion and have tested the binding of additional TMDs for these proteins in the same construct background after having validated their correct membrane integration. Our experimental results do not fully agree with the predicted values (Fig. 3c). We attribute this to the limited dataset used for algorithm training and think that further development of the algorithm and more extensive training would be needed to more accurately predict affinities for isolated TMDs within a single protein, whereas binding/no binding seems to be well predicted. To reflect this discrepancy, we have revised our wording in the manuscript results section and decided to exclude all prediction data from the revision that depends on the affinity prediction of a single TMD within a protein in comparison to other TMDs within the same protein. We also added a paragraph in the discussion section that specifically points out the current limitations of the algorithm.

4. Is the binding interface with the EMC1 TMD similar in the MD simulations with SLC4A2 TMD3 and TM4SF1 TMD4 to that shown in Figure 4b? It would be informative to show molecular snapshots of the MD simulations for these natural substrates and the strong binders in Figure 5 as further evidence for the designated chaperone interaction site.

We have included snapshots of representative poses of the MD simulations in Supplementary Fig. 7c. To address this question in a quantitative manner, we performed a contact analysis for the MD simulations, similar to the one shown in Supplementary Fig. 5c. The results are now displayed in Supplementary Fig. 7d and suggest that generally similar residues are important for binding: particularly the residues S967, F970 and F974, which we also targeted in our mutagenesis (Fig. 4c), seemed to contact the different bound TMDs.

5. Line 486: the authors state that ConMem mutants D6, Y9 and R13 lead to “no TMD assembly”, but Extended Data Figure 7c shows that the CL tagged variants are still pulled-down with these variants considerably, suggesting the statement is too strong.

We fully agree that this was an overstatement and thus adjusted our wording to “not strongly assemble”.

The schematic in Figure 4b shows both GFP and CL variants are HA-tagged, therefore it should be clarified what the bands in the HA western blot in Extended Data Figure 7c represent. Do the bands around the 55 kDa marker in the CL western blot correspond to ConMem variant fusions formed due to no peptide skipping at the P2A sequence? If so, the authors should show that this band is present for all variants (e.g.

by increased exposure of the input) and not just for the D13 and D17 variants where different EMC binding/cell surface exposure is seen.

We appreciate the reviewer's careful observation and would like to offer an alternative explanation. We have observed that the two constructs in question are prone to form aggregates upon strong denaturation/heating, as exemplified by the test we have additionally conducted to address this point (see image below). As can be seen here, the bands suspected of being the result of failed ribosome skipping disappear when the samples are treated with high urea concentrations, instead forming a multitude of other species strongly resembling aggregated forms. We thus always use carefully optimized denaturing conditions for each membrane protein under analysis.

Figure 2: Denaturation of the samples shown in Supplementary Fig. 7e under different denaturing conditions. Samples were either treated as in Figure S7e (heating to 60°C for 10 min) or were additionally supplemented with 6 M urea. A band presumed to be indicative of failed P2A ribosome skipping has been marked with a red arrow. Upon urea treatment, this band disappears and instead a multitude of other bands resembling aggregated forms appear.

6. Line 499: again “other variants that did not assemble with R13 failed to promote transport to the cell surface” seems a strong statement given transport occurs to a lesser extent. The authors conclude that differential binding of ConMem variants to the EMC as assessed by *in vitro* co-IP (Figure 6b) is the reason for their differential cell-surface exposure (Figure 6c), however this correlation is rather indirect given a multitude of factors are likely at play in the trafficking of these constructs *in vivo*. To make the statement, “EMC binding to problematic transmembrane domains goes hand in hand with ER retention”, the authors should show supporting fluorescence microscopy for the strains in Figure 6.

As suggested, we have revised the wording in the main text. Additionally, we have performed immunofluorescence microscopy of all given transmembrane domain pairs as suggested. The obtained images support our hypothesis, that the respective ConMem-GFP variant is retained in the ER and co-expression of a cognate partner can promote transport to the cell surface (new Supplementary Figure 7g).

Reviewer #3 (Remarks to the Author):

The study investigates the ER membrane protein complex (EMC) as a chaperone for labile transmembrane domains (TMDs). The authors focused on the lipid-filled cavity of EMC, and its binding with various transmembrane proteins. Using interactomics, systematic mutagenesis, molecular dynamics simulations, and a machine learning tool (ipredEMC), the authors demonstrate that EMC binds TMDs via hydrophobic interactions, rectifies their orientation in the membrane, and facilitates assembly. The work is valuable in understanding EMC's dual roles. The integration of various techniques demonstrates a strength leading to the discovery. The work in general is of good quality. I have some comments for the authors' further consideration.

We thank reviewer #3 for the positive evaluation of our manuscript and the helpful suggestions. To address the reviewer's questions, we have included new data in the revision and have revised our text and figure legends to improve overall clarity and comprehensibility and to better describe experimental setups.

1. *It was found that three positions within EMC1 showed most crosslinks: at the brace/crossbar (R513), at the luminal side of the single EMC1 TMD (D961), and at its cytoplasmic side (R981). Further discussion can be provided on this finding. Why the three positions are specific.*

This question addresses data shown in Supplementary Fig. 1c. We see that multiple positions within the "brace" and the transmembrane domain of EMC1 yield moderate to high crosslinking efficiency as assessed by western blot. This measure is a result of multiple factors, such as Bpa incorporation efficiency, accessibility vs. burial inside the protein etc. As a result of this, positions 513, 960, 961, 980 and 981 display a higher crosslinking efficiency. In our structural model, these positions are well accessible and oriented towards what we hypothesize to be the chaperone client engagement site. Furthermore, some positions may be more prone to quenching by lipids. Additionally, in our simulations with ConMem R13 the residues that proved very effective in crosslinking are not the major substrate contact sites (whose mutation may reduce binding). And lastly, there are other positions showing a similar efficiency as the ones chosen for IP-MS. We limited ourselves to three positions and chose 913, 961 and 981 as these are well distributed across the lipid filled cavity.

2. *The authors employed ConMem to investigate the interaction between EMC and TMDs. It should better explain the reason of choosing the variants sites shown in Figure 2c to study the interaction.*

The selection was mostly due to practical reasons, to keep the amounts of experiments manageable, while at the same time covering at least part of the entire TMD.

3. The machine learning model should be explained better in detail. As the datasets for model training is not large, there can be problems like overfitting. Can the machine learning prediction be tested with the proteomic results shown in Figure 1c, to explain the significantly enriched transmembrane proteins?

As suggested, more details about the machine learning model have been added to the Methods section of the manuscript.

Addition to the methods:

A Multilayer Perceptron (MLP) regressor was used for prediction, as implemented in scikit-learn (version 1.6.1). The model was trained using the default settings, with the exception of the maximum number of iterations, which was set to `max_iter = 1000`. The network architecture consisted of a single hidden layer with 100 neurons and employed the rectified linear unit (ReLU) activation function. Optimization was performed using the Adam solver with an initial learning rate of 0.001. L2 regularization was applied with a coefficient of 0.0001.

Furthermore, as suggested we have now tested our algorithm on the proteomic results from Figure 1c and have found a consistency between the predictions and experimental data. These additions are reflected in the revised manuscript in a new Supplementary Figure 4b and additions to the results section.

Addition to the results:

To test the model's predictive performance on independent data, we evaluated its predictions on a subset of the proteomic data shown in Figure 1c. In a simplified approach, we always scored a multipass TM protein with the highest score found for one of its TMDs (see Methods for details). We found that proteins identified as EMC binders by mass spectrometry indeed exhibited higher predicted EMC affinity scores (Supplementary Fig. 4b).

4. The authors note that ~50% of top ipredEMC hits are transporters (Fig. 3e) but do not discuss why this class is particularly reliant on EMC. Are their TMDs enriched in polar residues or other instability features?

Indeed, transporters are often enriched in polar/charged residues. We now have added some sentences on this in the revised manuscript.

5. In Fig. 4e, the absence of crosslinks in wild-type EMC1 controls is not quantified. A negative control (e.g., non-binding TMD) would strengthen this experiment.

We apologize for the confusion the data presentation has caused here. To clarify our data presentation, we have now added arrows pointing directly to the crosslinks and moved the input and mock data to the SI. The EMC WT (non-cysteine) and ConMem R13 (also non-cysteine) are our negative controls in this experiment.

6. The study posits that EMC binding “rectifies” TMD orientation (Fig. 5), but the functional consequence of this reorientation (e.g., enhanced folding or assembly) is not directly tested.

A direct test would be very difficult, but we have extended our interpretation of this in the revised discussion now.

7. The link between EMC’s insertase and chaperone functions remains speculative. Competition experiments (e.g., co-expressing insertase and chaperone clients) could address whether these roles are mutually exclusive or synergistic.

This indeed is an excellent point, which we now more extensively discuss in the revised manuscript, also based on the little literature available (which argues for synergy within a client, but competition between clients). Experimentally addressing this point, we consider to be more suitable for future studies that focus on the link between the different EMC functions, which now has become very relevant also based on our study.

8. As the work is rather complicated with the combination of various techniques, a summary of the main finding at the end of the discussion section is valuable.

This is a very good suggestion, and, at the beginning of the discussion, we now provide an extended summary of our findings.

Reviewer #3 (Remarks on code availability):

The codes are clear.

First and foremost, we would like to thank all three reviewers for the time they have invested in re-reviewing our manuscript. In the following, please find a detailed reply to the remaining points that have been raised during the re-review process:

Reviewer #1 (Remarks to the Author):

The authors have addressed my concerns, and I have no concerns.

Reviewer #2 (Remarks to the Author):

The revised manuscript by Klose et al., contains several additional control experiments and further analyses in response to the reviewer's comments. As a result, the data are compelling, significantly advance our understanding of the EMC and are ready for publication. I have a few minor comments that would improve clarity/interpretability for the reader:

1. I would explicitly name the constructs that are misintegrated and excluded from further analyses in the legends of Supplementary Fig. 2c and 4e.

As suggested, we have now done this in the second revision.

2. For Supplementary Fig. 2c and 4e, the cytosolic glycosylation site would also give a shift upon PNGase treatment if the construct were misintegrated in the wrong topology, therefore the top panels are sufficient to reveal the misintegrated constructs. For simplification, I would suggest to remove the now-redundant bottom panels showing the results for the luminal glycosylation site. The cartoon showing misintegration in the wrong topology could be included as an alternative scenario in the top panel.

This is true, and we have now done this in the second revision.

3. On page 12, I would suggest to append a statement similar to "although the extent of binding does not directly correlate with the predicted affinities" to the sentence "we found that TMDs which ipredEMC predicted to bind to the EMC indeed did", as Fig. 3c is still designed to show the affinity predictions for the single TMDs of the multipass proteins.

We added a statement based on this valid suggestion.

4. On page 15, I would (perhaps in brackets) explicitly name the residues referred to by "the positions within EMC1 which were predicted to form the most contacts with residues of ConMem", "residues previously proposed to be important for chaperone client engagement by the EMC" and "we introduced mutations at the EMC1 brace" as it would really help with interpreting the data in Supplementary Fig. 5.

We have now added several residues explicitly in the revised text.

5. On page 16, I would change “while more distal mutants did less so” to “while more distal mutants did not” as Fig. 4d does not show a significant change in EMC binding for the T11 and T15 variants.

We changed the text based on this valid suggestion.

Reviewer #3 (Remarks to the Author):

My previous concerns have been addressed. I have no further questions. It would be better if the authors provide a highlighted version of the revised manuscript to illustrate the changes.

Reviewer #3 (Remarks on code availability):

The codes are clear.